# CLADDER: Assessing Causal Reasoning in Language Models

**Zhijing Jin**[1,2,*] **Yuen Chen**[1,*] **Felix Leeb**[1,*] **Luigi Gresele**[1,*]

**Ojasv Kamal**[3], **Zhiheng Lyu**[4], **Kevin Blin**[2], **Fernando Gonzalez**[2], **Max Kleiman-Weiner**[5],

**Mrinmaya Sachan**[2], **Bernhard Schölkopf**[1]

[1]MPI for Intelligent Systems, Tübingen   [2]ETH Zürich   [3]IIT Kharagpur
[4]University of Hong Kong   [5]University of Washington

```
jinzhi@ethz.ch    chenyuen0103@berkeley.edu
fleeb@tue.mpg.de    luigi.gresele@tue.mpg.de
```

## Abstract

The ability to perform causal reasoning is widely considered a core feature of intelligence. In this work, we investigate whether large language models (LLMs) can coherently reason about causality. Much of the existing work in natural language processing (NLP) focuses on evaluating *commonsense* causal reasoning in LLMs, thus failing to assess whether a model can perform causal inference in accordance with a set of well-defined *formal rules*. To address this, we propose a new NLP task, *causal inference in natural language*, inspired by the *"causal inference engine"* postulated by Judea Pearl et al. We compose a large dataset, CLADDER, with 10K samples: based on a collection of causal graphs and queries (associational, interventional, and counterfactual), we obtain symbolic questions and ground-truth answers, through an oracle causal inference engine. These are then translated into natural language. We evaluate multiple LLMs on our dataset, and we introduce and evaluate a bespoke chain-of-thought prompting strategy, CAUSALCOT. We show that our task is highly challenging for LLMs, and we conduct an in-depth analysis to gain deeper insights into the causal reasoning abilities of LLMs.[1]

## 1 Introduction

> *Once we really understand the logic behind causal thinking, we could emulate it on modern computers and create an "artificial scientist".*
>
> — *Pearl and Mackenzie [2018]*

Causal reasoning is believed to be one of the hallmarks of human intelligence [29, 68]. The ability to draw causal inferences from available information is crucial for scientific understanding and rational decision-making: for example, knowing whether smoking causes cancer might enable consumers to make a more informed decision [17, 18]; assessing the causal effect of a vaccine is essential for effective policy-making during a pandemic [14, 44, 72, 97]; and understanding the interplay behind family background, education and income helps devise effective education policies [10, 11, 30, 73].

Our opening quote therefore mirrors the aspirations of many scientists in artificial intelligence and causal inference: to construct a machine capable of performing sound causal reasoning, and able to answer causal questions at scale and with ease. Recent advances in large language models (LLMs) have brought about a paradigm shift in natural language processing (NLP) and artificial intelligence [7, 15, 39, 56, 76, 103, *inter alia*]. These transformative developments raise the question of whether these machines are already capable of causal reasoning: *Do LLMs understand causality?*

---

[*]Main contributors.

[1]Our data is open-sourced at `https://huggingface.co/datasets/causalNLP/cladder`, and our code can be found at `https://github.com/causalNLP/cladder`.

37th Conference on Neural Information Processing Systems (NeurIPS 2023).

**Question:** Imagine a self-contained, hypothetical world with only the following conditions, and without any unmentioned factors or causal relationships:

**Physical vulnerability** has a direct effect on the likelihood of **fatality** and **vaccination decision**. **Vaccination** has a direct effect on the **fatality rate**.

In the entire population, 50% of the people are vulnerable to a certain disease.
For vulnerable and vaccinated people, the fatality rate is 4%. For vulnerable and unvaccinated people, the fatality rate is 7%.
For strong and vaccinated people, the fatality rate is 1%. For strong and unvaccinated people, the fatality rate is 5.8%.
Overall, the fatality rate for vaccinated people is 5%, while the fatality rate for unvaccinated people is 4.5%.

*Does getting vaccinated increase the likelihood of death?*

**CLadder**

**Ground-Truth Answer:** No

**Correct steps to lead to the ground-truth answer:**

**1) Parse the** *causal graph*: Confounding
Subskill: Causal Relation Extraction

**2) Classify the** *query type*: Average Treatment Effect
Subskill: Causal Question Classification

**3) Formulate the query to its** *symbolic form*:
$E[Y \mid do(X=1)] - E[Y \mid do(X=0)]$
Subskill: Formalization

**4) Collect the** *available data*:
$P(Z=1)=0.5$
Subskill: Semantic Parsing
$P(Y=1 \mid Z=1,X=1)=0.04, P(Y=1 \mid Z=1,X=0)=0.07$
$P(Y=1 \mid Z=0,X=1)=0.01, P(Y=1 \mid Z=0,X=0)=0.058$
$P(Y=1 \mid X=1)=0.05, P(Y=1 \mid X=0)=0.045$

**5) Derive the estimand** using *causal inference*:
$E[Y \mid do(X=1)] - E[Y \mid do(X=0)]$
Subskill: Formal Causal Inference
$= \sum_{Z=v} P(Z=z)*[P(Y=1 \mid Z=z,X=1) - P(Y=1 \mid Z=z, X=0)]$ # remove "do" using do-calculus
$= P(Z=0)*[P(Y=1 \mid Z=0,X=1) - P(Y=1 \mid Z=0,X=0)]$
$+ P(Z=1)*[P(Y=1 \mid Z=1,X=1) - P(Y=1 \mid Z=1,X=0)]$ # turn the expression into terms in the available data

**6) Solve for** the estimand by plugging in the relevant data in Step 4:
$= 0.5*(0.01 - 0.058)+0.5*(0.04-0.07)$ # plug in the numbers in the available data
Subskill: Arithmetics
$= -0.039$
$< 0$ # the effect size is negative, so the final answer is "No"

Figure 1: Example question in our CLADDER dataset featuring an instance of *Simpson's paradox* [63]. We generate the following (symbolic) triple: (i) the causal query; (ii) the ground-truth answer, derived through a *causal inference engine* [66]; and (iii) a step-by-step explanation. We then *verbalize* these questions by turning them into stories, inspired by examples from the causality literature, which can be expressed in natural language.

Many previous works addressed the above question by focusing on *commonsense* causality [34, 100, 101], inspired by the literature that explores LLMs as *knowledge bases* [40, 70, 83] (we refer to this line of work as *causality as knowledge*). This involves assessing the alignment between commonsense knowledge about causal relationships in humans and LLMs. This line of work generally does not focus on evaluating how well models are capable of *causal reasoning*. For example, it may be difficult to rule out the possibility that LLMs perform potentially unreliable *amortized causal inference*, answering causal questions by a simple repetition of verbal patterns present in the texts composing their training data:[2,3] in other words, LLMs may just be *"causal parrots"* [100].

In this work, we introduce a way to test the *formal causal reasoning in LLMs*. To this end, we introduce the CLADDER dataset. The specificity of CLADDER is that causal questions posed in natural language are *grounded in symbolic questions and ground truth answers*: the latter are derived through an oracle *causal inference engine (CI engine)* [66], which abides by the rules of the causal inference approach described by Pearl [61], based on graphical models and structural causal models (SCMs) [23, 59, 61, 69, 88]. We compose more than 10,000 causal questions that cover a variety of causal queries across the three rungs of the *Ladder of Causation* [3, 66]—i.e., *associational (Rung 1)*, *interventional (Rung 2)*, and *counterfactual (Rung 3)*. We consider several causal graphs, giving rise to scenarios which require different causal inference abilities. Additionally, we generate ground-truth explanations with step-by-step reasoning for more in-depth analysis of LLM behavior. Our symbolic questions and answers are then *verbalized*, by turning them into stories which can be expressed in natural language. To probe whether LLMs employ amortized causal inference, we construct stories with commonsensical, as well as anti-commonsensical and with nonsensical causal relations: in these latter cases, amortized causal inference is expected to fail, whereas formal causal reasoning would still yield the correct answer. An example question from CLADDER is shown in Figure 1.

Exploiting CLADDER, we also introduce a method to elicit sound causal reasoning in LLMs and help them solve challenging causality questions. Specifically, we develop **CAUSALCOT**, a chain-of-thought prompting strategy [96] inspired by the CI engine, which prompts the LLM to extract the causal graph, causal query, and available "data" (e.g., conditional or interventional *do*-probabilities [24]) from the question, formalize them precisely, and perform correct causal inferences.

---

[2]which may itself contain instances of fallacious causal reasoning.

[3]The extent to which this would imply an inaptitude of LLMs for causal reasoning has been questioned [38].

Our experiments indicate that CAUSALCOT achieves an accuracy of 70.40%, which substantially improves the performance of vanilla GPT-4 by 8.37 points on CLADDER.

We summarize the *main contributions* of our work:

1. In contrast to most other works on causality in LLMs, focusing on *commonsense causal knowledge*, our goal is to assess the LLMs' ability to perform *formal causal reasoning* (briefly reviewed in Section 2).

2. We introduce CLADDER (Section 3), a dataset containing more than 10K causal questions, spanning all three rungs of the ladder of causation, several causal graphs, and various stories for verbalization.

3. We develop CAUSALCOT (Section 4), a chain-of-thought prompting strategy to elicit formal causal reasoning in LLMs, inspired by the *causal inference engine*.

4. We perform extensive experiments on eight LLMs (Section 5), analyze fine-grained errors to showcase the limitations of LLMs in formal causal reasoning, and suggest directions for future research.

## 2 Preliminaries on Causal Inference

Our dataset design takes inspiration from the *Causal Inference Engine* as postulated by Pearl and Mackenzie [66], see also [59]. We begin with a brief overview of the causality framework by Pearl et al. [67].[4] This framework was largely developed within the field of artificial intelligence, and therefore puts particular emphasis on *algorithmic* aspects of causal reasoning (e.g., [62])—which makes it particularly suited for our work, where we want to algorithmically generate ground truth answers to causal queries, without having to appeal to common sense to assess the correctness of an answer.

### 2.1 The Ladder of Causation

The *Ladder of Causation*, introduced by Pearl and Mackenzie [66], is a proposed taxonomy, and hierarchy, of causal inference tasks [3]. It consists of three distinct rungs.

**Rung 1 ("*seeing*").**    This describes statistical associations (*"How often do I take an aspirin when I have a headache?"*). Rung 1 deals with statistical dependences among random variables, and involves probabilistic reasoning about joint and conditional distributions, $P(X = x, Y = y)$ and $P(Y = y|X = x)$, which can be formalised through *Bayesian Networks* [12, 58] representing a set of variables and their conditional dependencies via a directed acyclic graph (DAG).

**Rung 2 ("*doing*").**    This enables us to formalize the concept of actively intervening in the world, and modifying it toward some end (*"If I take an aspirin now, will my headache subside?"*). Interventions can be formalized using the *do-operator* [24] and *Causal Bayesian Networks* [67] to represent, for example, the distribution over $Y$ when intervening on $X$ to set its value to $x$ as $P(Y = y|\mathrm{do}(X = x))$.

**Rung 3 ("*imagining*").**    This rung deals with counterfactual reasoning, i.e., reasoning about alternative scenarios in which the world could have been different, possibly even contradicting the factual state (*"Would my headache have subsided, if I had taken an aspirin?"*). Counterfactual probabilities can be written as $P(Y_x = y)$, representing the probability that "$Y$ would be $y$, had $X$ been $x$". Reasoning about Rung 3 quantities requires the introduction of *Structural Causal Models (SCMs)* [67]. SCMs are especially powerful as they enable any quantity in Rungs 1, 2, and 3 to be formulated precisely [3].

### 2.2 Causal Inference

**Identification.**    Causal inference is especially difficult since we typically only have measurements from *lower* rungs, but want to reason about *higher* ones. A crucial question is then under what conditions are such inferences possible, i.e., what assumptions and measurements are required to unambiguously answer a causal query of interest: this is the question of *identification*. As argued in [3], *"it is generically impossible to draw higher-layer inferences using only lower-layer information"*. One may be able to draw inferences at a higher layer given a combination of partial knowledge of the underlying SCM, in the form of a causal graph, and data at lower layers. The graphical structure therefore plays a crucial role in bridging the rungs of the Ladder of Causation, and many prior works have been dedicated to exploiting properties of the graph to transform higher-rung queries into expressions which can be estimated based on lower-rung quantities [36, 64, 84].

---

[4]We refer to [3, 65] for a comprehensive introduction. See also Appendix C for further details.

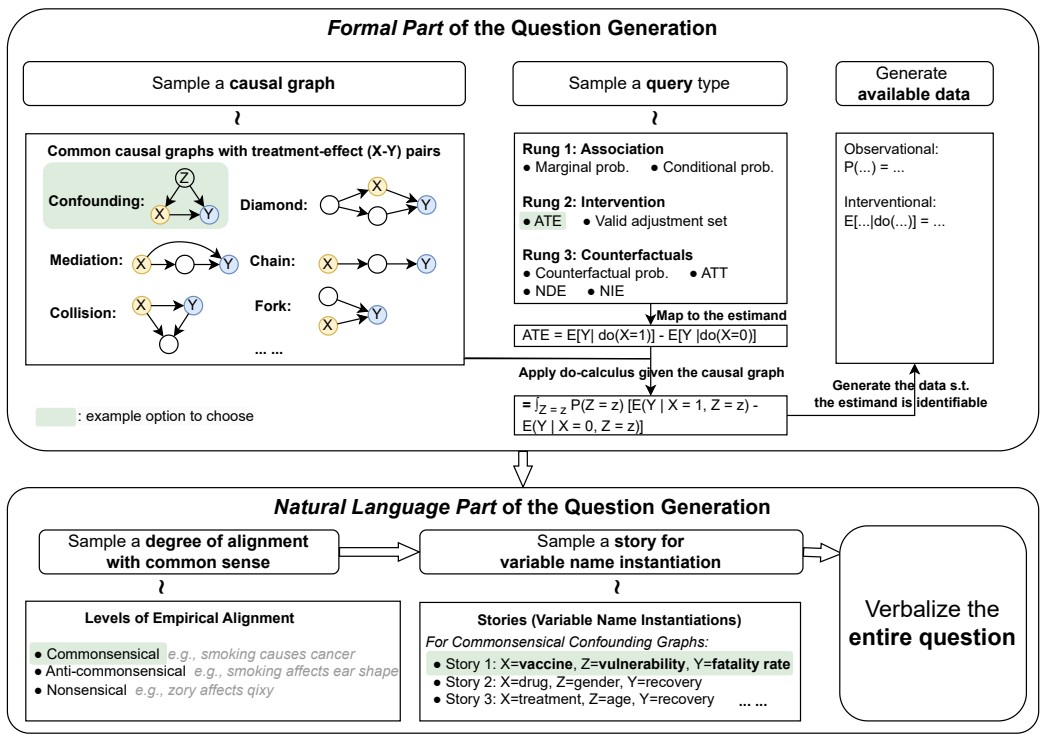

Figure 2: The data-generating process of the CLADDER dataset. The upper part of the figure describes the *formal part* of the question generation, which samples inputs for the CI Engine and derives a ground truth answer. The bottom part describes the *natural language part* of the question generation—i.e., its verbalization, based on multiple stories and different degrees of alignment with commonsense knowledge.

**Causal Inference Engine.** An overarching objective of this research is the construction of a *Causal Inference Engine (CI Engine)* [37, 59, 66], which takes as input a query, a graph, and some available data (typically from lower rungs than the query); and outputs whether a solution exists, and, if so, an equivalent expression of the query which is estimable from the available data. While some previous works refer to the CI engine in the context of Rung 2 queries, where it corresponds to the *do*-calculus [36, 84], here we refer to it in a more general sense, encompassing all three rungs.

## 3 Composing the CLADDER Dataset

**Task Formulation.** Like in the example of Figure 1, our dataset $\mathcal{D} := \{(q_i, a_i, e_i)\}_{i=1}^N$ consists of $N$ triples, each containing a question $q_i$, binary answer $a_i \in \{\text{Yes}, \text{No}\}$, and an explanation $e_i$. Our main task is to test the accuracy of the prediction function $f : q \mapsto a$, i.e., a LLM which maps a natural language causal question to an answer. Apart from directly evaluating the answer, we also compose the ground-truth explanations $e$ to evaluate the reasoning steps of LLMs.

**Design Principles.** In the composition of our dataset, we adhere to the following design principles. First, we ensure broad coverage of all rungs of the ladder of causation. Second, we avoid settings that involve continuous variables and use binary variables instead: this is partly due to the large availability of identifiability results for binary and categorical variables, and partly because queries involving binary variables lend themselves to more natural-sounding verbalization. Moreover, since LLMs struggle with calculation-heavy tasks [32, 91], and we are chiefly interested in causal reasoning abilities, we focus on graphs with few (three to four) variables, in various common configurations, to produce questions which are identifiable from the outset. Lastly, we carefully design a rich set of templates to translate the abstract formulas into grammatically correct and natural-sounding, fluent prompts.

**Overall Pipeline.** The generation pipeline for CLADDER, depicted in Figure 2, consists of two parts:

1. In the *Formal Part* (which we illustrate in Section 3.1), we specify all the required inputs (query, model, data) and the ground truth answer generated by the CI Engine.

2. In the *Natural Language Part* (in Section 3.2), we verbalize the formal queries and specification of the causal model and data by associating them to a story or narrative, using a rich set of templates.

## 3.1 Formal Part of the Question Formulation

The first step of our data generating process is to construct a set of inputs to the CI Engine such that *by design* there exists a well-defined ground truth answer: i.e., we construct triples of causal queries, graphs, and data such that the query can be unambiguously answered based on the available data (ensuring *identifiability* by construction).[5] The ground truth causal models, which specify all quantities which are considered measurable in our questions, are causal Bayesian networks (CBNs), where each causal mechanism (i.e., conditional probability of a variable given its parents in the factorization according to the causal graph $G$) corresponds to a Bernoulli distribution. We compile a selection of graphs $G$ based on examples drawn from multiple sources from the literature [66, 67, 69, 88], where suitable graph structures are used to illustrate toy problems in causal inference. The complete list of structures we consider can be found in Appendix A.3; the complete list of sources in Appendix A.1.

**Selecting Query Types.** We again draw from the causal inference literature to collect common query types in each rung. As illustrated in the *"Sample a query type"* box in Figure 2, for Rung 1, we can ask about probability distributions such as marginal probabilities and conditional probabilities. For Rung 2 questions, we can enquire *average treatment effects (ATE)* (*"how will $Y$ change if $X$ changes from $x$ to $x'$?"*), or what constitutes a valid adjustment set that can block all backdoor spurious correlations between $X$ and $Y$. Lastly, for Rung 3, we include *counterfactuals* (*"what would happen to $Y$ had $X$ been $x'$ instead of $x$?"*), *average treatment effect on the treated (ATT)* (*"for the subpopulation whose $X$ changed from $x$ to $x'$, how does their $Y$ change on average?"*), *natural direct effect (NDE)* (*"what is the direct effect of $X$ in $Y$, but not through the mediator?"*), and *natural indirect effect (NIE)* (*"what is the effect from $X$ to $Y$ through the mediator?"*).

**Applying the Causal Inference Engine for the Ground-truth answer.** By construction, the causal processes we define encapsulates all necessary information to make the causal quantities of the query types identifiable. This allows us to apply the rules of causal inference to obtain an estimand for each causal graph and query type, and evaluate the estimand to get a ground truth answer. The Rung 2 queries simplify to Rung 1 terms using the rules of *do*-calculus [59], and, for the Rung 3 queries, we apply methods of counterfactual causal inference [67] (with details in Appendix C.3). The estimand also specifies exactly which terms are necessary to include in the prompt as *"available data"* in order to ensure that enough information is provided to answer the question correctly (i.e., for identifiability), provided the correct causal reasoning is applied. Our entire code base of the data generation process can be found at our GitHub repository, `https://github.com/causalNLP/cladder`.

## 3.2 Natural Language Part of the Question Formulation

While Section 3.1 describes a way to generate the ground-truth causal model, query and answers, computed through a causal inference engine, real-world causal reasoning problems are expressed in natural language rather than symbolic expressions. The next part of the data generation pipeline therefore focuses on the verbalization of all these components with a plausible narrative in natural language.

**Generating the Stories.** For each causal graph, we collect a set of two to five *stories* which consist of a list of variable names for each node in the graph. The stories are primarily selected from examples in commonly cited causal inference books and papers (see Appendix A.1), which ensures that the stories and corresponding causal graph structures adhere to empirical common sense (e.g., the drug-gender-recovery example of Pearl and Mackenzie [66]). However, it is very likely that at least some of the stories appear in the training data of many LLMs. Therefore, we also generate various *anti-common sense* and *nonsensical* variants of the stories, meant to isolate the effects of memorization. For the anti-commonsensical stories, we randomly do one of the actions: (1) replace the effect variable $Y$ with an unusual attribute, that would not be an effect variable in any of the stories (e.g., "ear shape"); or (2) create an irrelevant treatment variable $X$ that does not play a causal role in any of our commonsensical stories, such as "playing card games" (see Appendix A.7). For the nonsensical variants, we invent artificial words as variable names such as "zory" and "qixy" (see Appendix A.6). .

---

[5]We use the term "data" to denote numerical values of conditional or *do*-probabilities, and not as collections of data samples. This is in line with how the term is used in other descriptions of the CI Engine [37, 66].

**Verbalizing the Prompts.**  The verbalization procedure applies the mapping of symbolic variables to semantic concepts to form a plausible narrative for the underlying causal process and then translates the symbolic expressions from the underlying causal process to natural language using carefully designed templates.

Specifically, we use several different grammatical forms for each semantic concept $t$ in the story to make the resulting prompt sound natural and grammatically correct. We first have the overall variable name $v_{\mathrm{overall}}(t)$ (e.g., the recovery status), and, then, for each binary value $i \in \{0, 1\}$, we compose its noun $v_{\mathrm{noun}}(t = i)$ (e.g., recovery), verb (e.g., to recover), sentence $v_{\mathrm{sent}}(t = i)$ (e.g., the patients recover), noun with attributive clause $v_{\mathrm{attr}}(t = i)$ (e.g., patients who recover), and third conditional $v_{\mathrm{cond}}(t = i)$ (e.g., if the patient had recovered).

Using these elements, we first verbalize the causal graph by iterating through each node and its outgoing edges, using the template "$t$ has a direct effect on $\mathbf{CH}(t)$.", where $\mathbf{CH}(\cdot)$ denotes the set of direct effects (children) of a variable. Then, for the available data $d$, we verbalize each conditional probability by "For $v_{\mathrm{attr}}(t_m = i)$, the probability of $v_{\mathrm{noun}}(t_n = 1)$ is $p$.", and each marginal probability by "The overall probability of $v_{\mathrm{attr}}(t = 1)$ is $p$." Note that our distributions are Bernoulli, so it is adequate to just introduce the parameter $p$, which is the likelihood of $t = 1$. For example, we generate sentences such as "The overall probability of recovery is 60%." and "For patients who have small kidney stones, the probability of recovery is 70%." Finally, for the query $q$, we instantiate each query type in our dataset following our question templates in Appendix A.5 such that the questions can always be answered with "yes" or "no".

**Generating the Explanations.**  Apart from the question-answer pairs, we also generate the step-by-step explanations. Our goal is to provide all intermediate reasoning steps a student of causal inference would use to answer the questions, so that each necessary subskill necessary for causal inference can be evaluated individually. We identify the following six subskills: ① causal graph extraction; ② correct query type interpretation; ③ symbolic formalization of the query; ④ semantic parsing to compile the available data; ⑤ estimand derivation; and ⑥ arithmetic calculation to solve the estimand, as in the colored boxes in Figure 1. Our explanation $e$ verbalizes all the elements ①-⑥ as sequential steps using our template in Appendix A.8.

## 3.3 Dataset Statistics

Our data-generating procedure has the potential to algorithmically generate a vast large number of questions. In practice, we pick a dataset size that is large enough to be representative, and at the same time not too large to be problematic given the expensive inference costs of LLMs. We therefore set our dataset size to be 10K, and report the statistics in Table 1.

The dataset roughly balance across the query types, graph structures, stories, and ground truth answers (as seen in Figure 3). Note that some causal queries are only compatible with a subset of the graphs, thereby resulting in a slightly lower representation of those queries (such as the NDE and NIE). More details on our design choices can be found in Appendix A.4.

|  | Total | Rung 1 | Rung 2 | Rung 3 |
|---|---|---|---|---|
| **Size** | | | | |
| # Samples | 10,112 | 3,160 | 3,160 | 3,792 |
| **Question** | | | | |
| # Sentences/Sample | 6.01 | 5.88 | 5.37 | 6.65 |
| # Words/Sample | 80.9 | 73.43 | 76.95 | 90.42 |
| # Nodes/Graph | 3.52 | 3.5 | 3.5 | 3.54 |
| # Edges/Graph | 3.38 | 3.3 | 3.3 | 3.5 |
| **Answer** | | | | |
| Positive Class (%) | 50 | 50 | 50 | 50 |
| **Explanations** | | | | |
| # Sentences/Sample | 9.11 | 9.1 | 8.1 | 9.96 |
| # Words/Sample | 47.95 | 49.87 | 32.8 | 58.97 |

Table 1: Statistics of our CLADDER dataset v1.5.

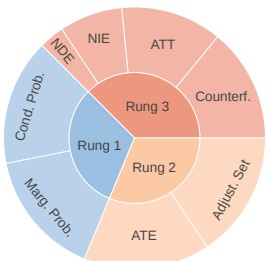

Figure 3: Distributions of query types in our 10K data.

## 3.4 Data Quality Check

Our dataset is generated through an algorithmic procedure, which has the following potential benefits: formal correctness; zero human annotation cost; and, most importantly, controllability—e.g., for

the question distribution, as well as for making it more unlikely that the data was previously seen by the model. However, since the dataset is different from common NLP datasets collected from human natural language writing, we also need to perform additional data quality checks. We therefore checked for a list of non-formal, natural language properties: grammaticality; human readability; naturalness/perplexity; and how well humans perform on this task.

For grammaticality, we ran a grammatical error check on our dataset using the LanguageTool package [51], and got on average 1.26 grammatical errors per 100 words (i.e., 98.74% correctness), which shows that most of the language in our dataset follows English grammar. For human readability, we checked how comprehensible the questions are to students who have taken causality courses. We selected a random subset of 50 questions from the dataset, and let a graduate student annotator go through the questions to judge whether they could understand them or not: 96% of the questions were deemed readable. Next, for the naturalness/perplexity score, we used the open-sourced GPT-2 model and obtained a perplexity score of 21.17 on our dataset, which is substantially lower (i.e., closer to the distribution of natural human-written text) than the one of MATH [32], a commonly used dataset of maths questions. Lastly, we conducted a sanity check where one expert evaluator tried to solve a random sample of 50 questions from the dataset, and we recorded an accuracy of 82% on this task.

# 4 Our CAUSALCOT Model

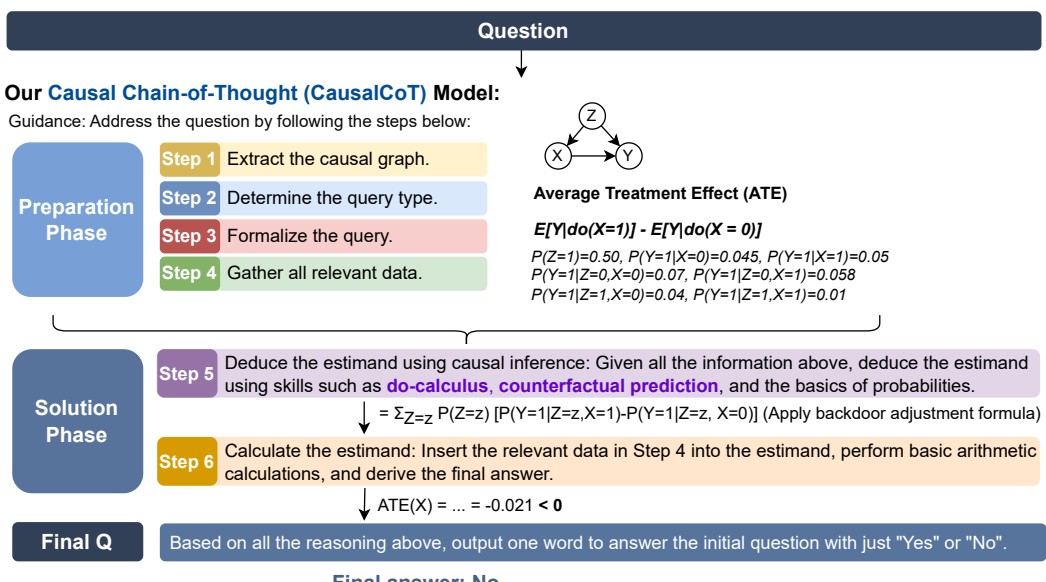

Figure 4: Illustration of our CAUSALCOT prompting strategy, which designs a chain of subquestions inspired by the idea of a CI engine [66].

In order to guide LLMs in correctly answering the questions in CLADDER, we draw inspiration from the ideal functioning of the CI engine [66], which breaks down a causal reasoning problem into multiple symbolically-grounded, simpler steps. We develop CAUSALCOT, a multi-step causal chain-of-thought prompt in Figure 4, which combines formal causal reasoning skills with the idea of chain-of-thought prompting [96] and the use of scratch pads for solving more complicated problems requiring a long list of steps [55] for LLMs.

We base our prompt design on the multi-step reasoning process of causal inference as shown in Figure 4, first starting with four preparation steps: ① identifying the causal graph structure; ② determining the causal query type;[6] ③ formulating the query symbolically precisely; and ④ extracting relevant data from the prompt. Then, given all the information collected in the preparation stage, we introduce the formal solution: ⑤ correctly deducing the estimand using causal inference techniques; and finally ⑥ evaluating the estimand to answer the question. This set of steps require both *natural language understanding* to parse the question (as in most steps in the preparation phase), as well as *formal causal reasoning* to derive the correct estimand (as in the solution phase).

---

[6]This step amounts to a multi-class classification problem, where each class is a different causal query.

We build our CAUSALCOT prompting strategy using GPT-4 [56], a recent autoregressive LLM that achieves state-of-the-art performance on many tasks. This latest model builds upon the previous series of general pretrained models (GPT) [7, 76] and adds reinforcement learning with human feedback, or instruction-tuning [1, 57, 104], to align the model responses to free-form questions with human preferences. It has achieved human-competitive performance over a list of tasks [8, 43, 54, 56, 105], among which the more formal tasks unseen in the training data still remain elusive [42, 78, 91].

Given a causal question $q$, we provide the LLM a list of instructions $\ell := (s_1, \ldots, s_6)$ consisting of the detailed descriptions of the six steps $s_1, \ldots, s_6$ in Figure 4. As the model $f_{\text{LLM}} : s_i \mapsto r_i$ autoregressively produces responses $r_1, \cdots, r_6$ sequentially corresponding to the six steps, we concatenate all the above before asking the final question "Based on all the reasoning above, output one word to answer the initial question with just 'Yes' or 'No'." See the complete prompt in Appendix B.1. In the end, we obtain the binary answer $a \in \{\text{Yes}, \text{No}\}$ as the final result.

Compared with the standard strategy of directly prompting the LLMs a question, we impose an *inductive bias* upon LLMs by using the causal inference framework, thus incorporating some of the powerful, principled insights of the causal inference community for NLP tasks. In this way, we enhance the strong natural language ability of LLMs with formal causal reasoning skills.

## 5 Testing LLMs with CLADDER

### 5.1 Experimental Setup

Our empirical investigation focuses on some of the most recent language models. We include the latest GPT-4 [56] with 1T parameters by the time we conduct the experiments (i.e., gpt-4-1106-preview), the previous ChatGPT (i.e., GPT-3.5) with 175B parameters, and then a series of earlier models with instruction-tuning on the 175B GPT-3 (text-davinci-001, -002, and -003) [57]. As baselines, we also include the non-instruction-tuned GPT-3 (davinci). We use the OpenAI API with temperature 0 when querying these models. We also include open-source, more efficient models like LLaMa [93] and its instruction-tuned version Alpaca [92], both with the same number of parameters, 6.7B.

### 5.2 Main Results

| | Overall Acc. | Acc. by Rung | | | Acc. by Commonsense Alignment | | |
| --- | --- | --- | --- | --- | --- | --- | --- |
| | | 1 | 2 | 3 | Comm. | Nonsens. | Anti-C. |
| Random | 49.27 | 50.28 | 48.40 | 49.12 | 49.01 | 49.69 | 49.12 |
| LLaMa | 44.03 | 48.23 | 29.46 | 52.66 | 45.14 | 44.22 | 42.67 |
| Alpaca | 44.66 | 52.03 | 29.53 | 51.13 | 44.86 | 44.40 | 44.77 |
| GPT-3 Non-Instr. (davinci) | 49.92 | 50.00 | 49.75 | 50.00 | 49.06 | 49.97 | 50.72 |
| GPT-3 Instr. (text-davinci-001) | 51.40 | 51.30 | 52.63 | 50.47 | 54.31 | 50.13 | 50.05 |
| GPT-3 Instr. (text-davinci-002) | 53.15 | 50.85 | 56.96 | 51.90 | 55.33 | 52.47 | 51.81 |
| GPT-3 Instr. (text-davinci-003) | 56.26 | 51.11 | 62.97 | 54.96 | 56.83 | 54.79 | 57.49 |
| GPT-3.5 | 52.18 | 51.80 | 54.78 | 50.32 | 54.09 | 50.68 | 52.09 |
| GPT-4 | 62.03 | 63.01 | 62.82 | 60.55 | 62.27 | 63.09 | 60.47 |
| + CAUSALCOT | **70.40** | **83.35** | **67.47** | **62.05** | **69.25** | **71.58** | **70.12** |

Table 2: Performance of all models on our CLADDER dataset v1.5. We report the overall accuracy (Acc.), and also fine-grained accuracy by rung, and by degree of commonsense alignment, from commonsensical (Comm.), nonsensical (Nonsens.), to anti-commonsensical (Anti-C.).

We compare the performance of all models in Table 2. First, we can see that the causal reasoning task in CLADDER is in general very challenging for all models. Models such as the earlier, non-instruction-tuned GPT-3, and both LLaMa and Alpaca are around random performance. With instruction-tuning, models start to show some improvement. And amongst all, our CAUSALCOT achieves the highest performance of 70.40%, which is substantially better than the vanilla GPT-4 by 8.37 points. Moreover, CAUSALCOT also achieve the best performance across all three rungs of causal questions, with a monotonically decreasing performance as the rungs get higher, i.e., the questions get more difficult. See Appendix D for experiments on our earlier dataset v1.0.

### 5.3 Isolating the Effect of Data Contamination

A well-known problem with evaluating LLMs on question-answering tasks is the data contamination problem, i.e., that LLMs perform well on a test set because the test set is (unintentionally) contained partially or even entirely in the training data [7, 56]. We address this problem by creating not only the commonsensical subset of our dataset, but also anti-commonsensical and nonsensical, both of which,

by construction, are very likely not in the training data of LLMs. From the accuracy by commonsense alignment degree in Table 2, we can see the original GPT-4 model performs the worst on the anti-commonsensical subset (1.8 points lower than that on the commonsensical subset). However, our CAUSALCOT enhances the reasoning ability across all levels, with substantial improvement on anti-commonsensical data by 9.65 points, highlighting the strength of CAUSALCOT on unseen data.

## 5.4 Error Analysis by Subquestions

| Step ① | | | Step ② | | | | Step ③ & ⑤ | Step ④ | Step ⑥ |
|---|---|---|---|---|---|---|---|---|---|
| Node | Edge | Dist. (↓) | Overall F1 | Rung 1 | Rung 2 | Rung 3 | Estimand | F1 | Arithmetic |
| 99.34 | 97.01 | 1.69 | 50.65 | 69.99 | 59.14 | 42.12 | 53 | 47.53 | 99 |

Table 3: Performance for each step in CAUSALCOT. For Step ①, we report the F1 score of node prediction, edge prediction, and also the graph edit distance (Dist.) with the true graph. See more details in Appendix E.1.

We conduct a fine-grained error analysis by looking into the performance of different steps of CAUSALCOT in Table 3.[7] We can see that the model is good at Step ① to extract causal graph $\mathcal{G}$, achieving high F1 scores for predicting both the nodes and the edges correctly, although not perfect, still leaving a graph edit distance of 1.69 between the ground truth causal graph and the model-identified graph. The other steps are more challenging for the model. Among those, Steps ②, ③ and ⑤ require careful and correct application of causal inference, where the model struggles. This reveals a notable weakness of current LLMs to perform formal causal reasoning, which is an important direction for future work on improving and enhancing LLMs. To better understand the reasoning abilities of LLMs, we also perform an extensive analysis taking the entire reasoning chain of our CAUSALCOT and the ground-truth explanations, to produce 20 fine-grained scores about the multi-step reasoning quality using the ROSCOE framework [25], and show detailed results in Appendix E.2.

## 5.5 Effect of In-Context Learning

As an additional analysis, we look into the effect of in-context learning (ICL) by providing an example solution before asking the question. The interesting question to us is whether models can generalize across different query types. Namely, we keep our CAUSALCOT framework, and prepend a reasoning example of query type $i$, and then calculate how much improvement it can bring when models answer new questions of query type $j$. In Figure 5, we can see that conditional probability and NIE are the questions that benefit the most from ICL, and showing examples of marginal probability and ATT are among the most helpful to all questions in general.

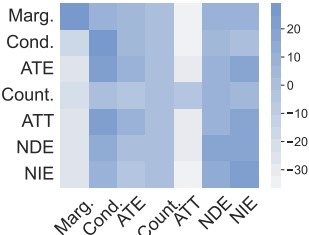

Figure 5: Heatmap showing the how helpful each query type is to solving subsequent query types.

## 6 Related Work

**Skill evaluation for LLMs.** Our work may be seen as part of the literature aimed at evaluating the performance of current LLMs [7, 15, 56, 76, 103, *inter alia*], focusing on understanding their strengths and weaknesses. Various studies into the capabilities of LLMs [8, 39, 56, 74] change people's perception of domains such as education [2, 80], medicine [54, 87], law [43], and computational social science [105]. However, most work evaluates new models on existing datasets from previously-curated large-scale benchmarks [89, 94, 95], or human exams [41, 43, 56] which is becoming increasingly unreliable due to training set contamination.

**Causality-related skills for NLP.** With the increasing attention on LLMs and causality [100, 101], we review several formulations of causality-related skills for NLP, which we summarize into (1) causality as knowledge, (2) causality as language comprehension, and (3) causality as reasoning. In the *causality-as-knowledge* line of work, many existing studies investigate how well NLP models understand commonsense causality, such as the cause and effect of an agent's action [81], motivation and emotional reaction in a social context [82], correspondence of a set of steps with a high-level goal [102], development of a story given a different beginning [75], and how in general LLMs serve as a knowledge base of causality [100]. Concurrent work [45] focuses on evaluating LLMs on various causality related tasks by leveraging the conceptual knowledge accrued from the training

---

[7]We experienced some rate-limiting in the fine-grained analysis of LLMs that are only accessible through a web API. As a result, we occasionally had to evaluate on a subset of 2K random samples.

data, rather than formal causal inference, except for their causal sufficiency analysis which is close to our counterfactual questions. Importantly, most work in this line does not define explicit causal graphs, making it difficult to quantitatively define the ground-truth causal relationships in a principled way. The *causality-as-language-comprehension* line of work stems from traditional linguistic studies on causal connectives and causal language usage [9, 90, 99], to the recent causal relation extraction [4, 33, 98] to identify cause-effect pairs as a subtask of information extraction from text.

Finally, for *causality as formal reasoning*, our CLADDER work formulates the task of causal inference for NLP, and our other work, CORR2CAUSE [42], addresses the causal discovery problem to infer causation from correlation. Together, they cover the two major branches of causal reasoning investigated in existing technical literature on causality. See a comprehensive comparison of literature in Appendix F.

## 7 Discussion of Limitations and Future Work

**A Natural Language *"Mini Turing Test"* for Causality.** Pearl and Mackenzie [66] describe an ideal *"mini-Turing test"* to assess understanding of causal inference, and argue that if a machine can answer all possible questions correctly, then it "understands" causality. According to the authors, this is because there are no possible shortcuts when you consider all possible combinations of queries, graphs and data in this ideal test: due to their combinatorial explosion, the machine can only answer all questions right if it correctly applies causal reasoning. From this point of view, our work constitutes a *first step towards a mini-Turing test formulated in natural language*. However, we cover only some of the commonly studied causal queries spanning all three rungs. Future work may extend this to further queries, such as, e.g., path-specific effects other than NDE and NIE [52], thereby increasing the number of potential questions and moving closer to the ideal test.

**LLMs and Causal Reasoning.** It has been claimed that LLMs understand causality well (e.g., [45] report high performance, such as 97% and 92%). In contrast, our work suggests that LLMs may still be far from reasoning reliably about causality (reaching only 60+% on CLADDER). As argued in Section 1, we believe that investigating this aspect may be of particular importance, since causal inference is crucial in many policy-relevant scenarios, where reliable AI systems could assist decision-making: from epidemiology [22, 79] to economics [10, 37] to fairness [47, 71]. Testing the abilities of these systems in semi-realistic scenarios is therefore crucial, motivating some of the design choices in our dataset: e.g., the example in Figure 1 was inspired by similar questions which arose in the context of the COVID-19 pandemic, where incorrect causal reasoning resulted in a fallacy where vaccinations were considered to be harmful instead of beneficial [20, 49]. Further work may be dedicated to making the questions and verbalizations even closer to realistic instances of causal inference problems.

**A CI Engine Plug-in for LLMs.** An interesting direction for future research could be to provide the LLM access to an actual implementation of the CI engine. For example, Davis and Aaronson [13] tested the improvement of math abilities in LLMs augmented with plug-ins (i.e., external modules that extend the model's capabilities by adding specific functionality or customizing its behaviour for particular tasks, like a calculator), suggesting that they significantly enhance the model's ability to solve these problems. However, even with plug-ins, there are still often *"interface"* failures: that is, *"[the LLM] often has trouble formulating problems in a way that elicits useful answers from the plug-ins"*. We hypothesise that something similar would happen for causal inference: even once suitable plug-ins are built, the language-to-tool interface may still be a non-trivial research question.

## 8 Conclusion

We proposed formal causal reasoning as a new task to evaluate LLMs, and created the CLADDER benchmark, covering several aspects of causal inference across all rungs of the ladder of causation and verbalizations involving semi-realistic scenarios. To address the task, we proposed a prompting strategy, CAUSALCOT, inspired by the principles of formal causal inference, which introduces multistep chain-of-thought reasoning for causal questions. Extensive experiments indicate that this dataset is highly challenging, thus offering a principled tool to gain a better understanding of the reasoning abilities of LLMs and to develop better models for causal reasoning in natural language.

## Acknowledgment

We thank Sergio Hernan Garrido Mejia for pointing us to Python implementations of the causal inference engine. We thank Armen Aghajanyan for the idea of testing psychological bias in LLMs,

which partly contributes to the idea of exposing causal bias in LLMs. We thank András Strausz for various timely coding help, especially for our Simpson's paradox case.

The material presented in this manuscript is partly based upon works supported by the German Federal Ministry of Education and Research (BMBF): Tübingen AI Center, FKZ: 01IS18039B; by the Machine Learning Cluster of Excellence, EXC number 2064/1 – Project number 390727645; the Swiss National Science Foundation (Project No. 197155); a Responsible AI grant by the Haslerstiftung; an ETH Grant (ETH-19 21-1); and by the John Templeton Foundation (grant #61156). Zhijing Jin is supported by PhD fellowships from the Future of Life Institute and Open Philanthropy, as well as the travel support from ELISE (GA no 951847) for the ELLIS program. Felix Leeb is supported by the International Max Planck Research School for Intelligent Systems (IMPRS-IS). Luigi Gresele is supported by the VideoPredict project, FKZ: 01IS21088.

## Author Contributions

The conceptualization and design of this project was led by Zhijing, Luigi and Felix, and supervised by Mrinmaya on the NLP part, and Bernhard on the causality part. Max provided timely insights from cognitive science on different types of causal tasks and on the project design. In the exploration stage, Ojasv did substantial work on discovering causal fallacies in news and on Twitter, which, while not included in the current systematic way of generating causal inference questions, was a significant contribution in the course of the project and in comparing various task formulations.

As for the operationalization and programming, the dataset composition was mainly led by Yuen and Felix, together with daily discussions with Zhijing, and weekly discussions with Luigi. Zhiheng supported an important function of generating the backdoor adjustment set for a given causal graph with the treatment and effect variables. The experiments are mainly conducted by Zhijing and Fernando, with Kevin finishing the evaluation results using the ROSCOE package.

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

# A  Supplementary for Dataset Generation

## A.1  List of References for Causal Inference

When collecting the causal graphs, query types, and commonsensical stories for our dataset, we took our examples from the following books (sorted by year):

1. Causality [67]
2. Causal inference in statistics: A Primer [23]
3. Elements of Causal Inference [69]
4. The Book of Why [66]
5. Introduction to Causal Inference [53]

And the following papers:

1. Causes and Explanations: A Structural-Model Approach. Part I: Causes [27]
2. Causes and Explanations: A Structural-Model Approach. Part II: Explanations [28]
3. Causality and Counterfactuals in the Situation Calculus [35]
4. Causal inference in statistics: An overview [60]

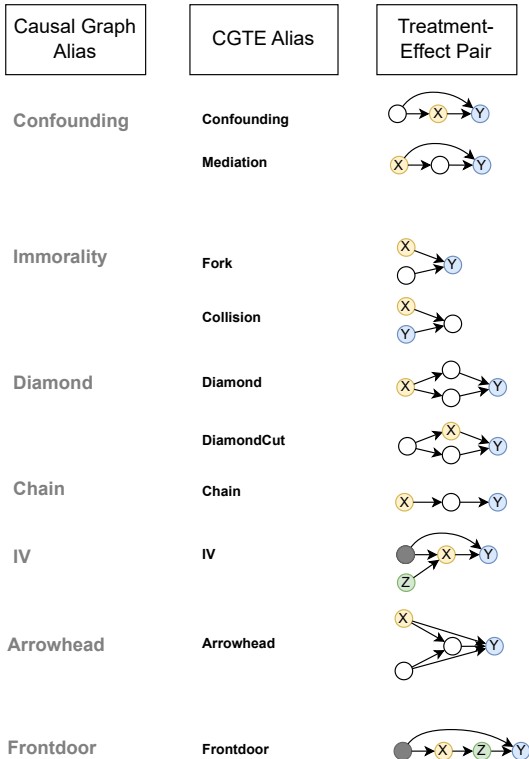

Figure 6: List of all ten causal graphs with treatment-effect pairs (CGTEs). We omit CGTEs that trivially resemble existing ones.

## A.2  Formulation of the Query Types

Here, we introduce all the query types included in our dataset.

**Rung-1 Queries: Marginal and Conditional Probabilities.**  For marginal probabilities, we ask questions about the overall distribution of a variable. For conditional probabilities, we ask whether conditioning on one variable increases or decreases the likelihood of another variable. For the explaining away questions, we condition on a collider node and ask how that affects the correlation between the two parents.

**Rung-2 Queries: ATE and Adjustment Set.** For ATE questions, we ask whether the treatment ($X = 1$) increases or decreases the likelihood of the effect variable $Y = y$. For adjustment set questions, we ask whether a set of variables should be adjusted for when estimating the causal effect between treatment and effect. By adjusting, we aim to blocked the non-causal paths from the treatments to effect, and hence eliminate spurious correlation. For example, to query whether the set gender is an adjustment set for the effect of a treatment on recovery, we ask *"To estimate the effect of the treatment on recovery, should we directly look at how the treatment correlates with recovery, or should we look at gender-specific correlation?"* In the collider bias questions, similarly to the explaining away questions, we condition on a collider variable and ask about how an intervention on one of the parents (treatment $X$) affects the other parent (outcome $Y$). However since by construction $X$ and $Y$ do not have common causes, the answer to this question is always "no".

**Rung-3 Queries: Counterfactual Probability, ATT, NDE, and NIE.** For counterfactual probability, we ask about what would have been the likelihood of $Y = y$, if the treatment variable $X$ had been $x$, given sufficient evidence $e$ such that the query is identifiable. For ATT, we ask how the likelihood of $Y = y$ would change for those who received treatment ($X = 1$) if there had been no treatment ($X = 0$). For NDE, we ask whether the $X = 1$ directly increases or decreases the likelihood of the $Y = y$, not through any mediators. For NIE, we ask whether the treatment (setting $X = 1$) increases or decreases the likelihood of $Y = y$ through mediators, not directly.

### A.3 Collection of Causal Graphs

We include all the ten causal graphs with treatment-effect pairs (CGTEs) in Figure 6.

Note that one causal graph can have several different CGTEs, such as the confounding structure, which has three CGTEs: confounding, mediation, and collision in the triangle form. To generate all the causal graphs and CGTEs here, we iterate all commonly used ones within four nodes in the CI books, and omit CGTEs whose solution by CI methods trivially resembles existing ones.

### A.4 Data Coverage

Starting from the full set of 12 distinct causal graphs and 10 query types, there are a few combinations that must be omitted as the ground truth answer would be trivial or ill-defined. For example, in the "Immorality" graph, the treatment "X" and outcome "Y" are by construction statistically independent, so there correlation is necessarily 0. Similarly, there are several graphs where certain causal queries are ill-defined or don't make sense to ask. Specifically:

1. For the Natural Direct Effect, we only include questions on the "IV", "Arrowhead", "Confounding", "Mediation" and "DiamondCut" graphs.
2. For the Natural Indirect Effect, we only include questions on the "Mediation", "Frontdoor", "Arrowhead", "Diamond" and "Chain" graphs.
3. For the Collider Bias and Explaining Away effect, we only include questions on the "Collision" graph.
4. For the Average Treatment Effect, we include questions on all graphs except "Collision".
5. For the (deterministic) Counterfactuals, we include questions on all graphs except "Collision".
6. For the Average Treatment Effect on the Treated (ATT), we include questions on all graphs except "Collision" and "IV".

The "balanced" benchmark (main benchmark in v1.5), containing 10,112 questions split between all stories, graphs, query types, and commonsensicalness, is balanced such that there are roughly the same number of questions for each distinct story-graph-query combination (ranging from 50-100 per combination) across the different variants: commonsense, anticommonsense, and nonsense. Furthermore, we balance the distribution of correct answers so that there are the same number of "yes"s and "no"s.

The "aggregate" variant (main benchmark in v1.0) contains 10,560 questions and is primarily balanced across all stories. However since the number of stories for each variant (commonsense, anticommonsense, and nonsense) varies significantly, the results in an unbalanced benchmark in terms of sensicalness.

## A.5 Query Form and Text Templates

We provide in Table 4 the text templates we use for each query type.

| Query Type | Symbolic Expression | Natural Language Question Template |
|---|---|---|
| **Rung 1: Association** | | |
| Marg. Prob. | $P(Y)$ | Is the overall likelihood of $\{v_{\text{noun}}(X = 1)\}$ greater than chance? |
| Cond. Prob. | $P(Y\|X)$ | Is the chance of $\{v_{\text{noun}}(Y = 1)\}$ larger when observing $\{v_{\text{noun}}(X = 1)\}$? |
| **Rung 2: Intervention** | | |
| ATE | $\mathbb{E}[Y\|do(X = 1)] - \mathbb{E}[Y\|do(X = 0)]$ | Will $\{v_{\text{noun}}(X = 1)\}$ increase the chance of $\{v_{\text{noun}}(Y = 1)\}$? |
| Adjust. Set | If $\boldsymbol{S}$ opens a backdoor path | To understand how $\{v_{\text{overall}}(X)\}$ affects $\{v_{\text{overall}}(Y = 1)\}$, should we look directly at how $\{v_{\text{overall}}(X)\}$ correlates with $\{v_{\text{overall}}(Y)\}$ in general, or this correlation case by case according to $\{v_{\text{overall}}(\boldsymbol{S})\}$? |
| **Rung 3: Counterfactuals** | | |
| Counterf. Prob. | $P(Y_x = y)$ | Can we infer that $\{v_{\text{sent}}(Y = 1)\}$ had it been that $\{v_{\text{cond}}(X = 1)\}$ instead of X=0? |
| ATT | $\mathbb{E}[Y_1 - Y_0\|X = 1]$ | For $\{v_{\text{attr}}(X = 1)\}$, would it be more likely to see $\{v_{\text{noun}}(Y = 1)\}$ $\{v_{\text{cond}}(X = 0)\}$? |
| NDE | $\mathbb{E}[Y_{1,M_0} - Y_{1,M_0}]$ | If we disregard the mediation effect through $\{v_{\text{overall}}(Y = 1)\}$, would $\{v_{\text{noun}}(X = 1)\}$ still positively affect $\{v_{\text{noun}}(Y = 1)\}$? |
| NIE | $\mathbb{E}[Y_{0,M_1} - Y_{0,M_0}]$ | Does $\{v_{\text{overall}}(X)\}$ affect $\{v_{\text{overall}}(Y)\}$ through $\{v_{\text{overall}}(\text{OtherVars})\}$? |

Table 4: Example natural language templates for each query type.

## A.6 Nonsensical Stories

To come up with a collection of nonsensical variable names, we use GPT-4 to generate some meaningless words. Specifically, we use the prompt: "Create 100 non-existent words that are short, i.e., within 5-characters.", with temperature=0 with the OpenAI interface. The collection of nonsensical words we later use as variable names are as follows: ziblo, truq, fyze, glimx, jorv, wexi, snov, yupt, kraz, qixy, vubr, chiz, pliv, moxa, fygo, rukz, tasp, xevo, jyke, wibl, zorf, quzy, nyrp, gwex, smez, vytz, hupx, cwoj, lirf, ovka, pexu, yigz, twaz, kwox, zuph, fraq, jyxo, swoy, uvzi, nekl, gyzp, rixq, vwem, xyfu, blyz, qwip, zeku, tijv, yomx, hwaz, czix, plof, muvy, fyqo, rujz, tasb, xevi, jyka, wibm, zorx, quzw, nyro, gwet, smeu, vyta, hupz, cwoi, lirg, ovki, pexy, yigw, twac, kwoz, zupj, fraq, jyxi, swoq, uvzo, nekm, gyzl, rixw, vwen, xyfo, blyx, qwiu, zeky, tijw, yomz, hwax, czir, ploz, muvq, fyqi, rujx, tasn, xevu, jyko, wibp, zory, and quzt.

## A.7 Anti-Commonsensical Stories

For the anti-commonsensical stories, we randomly do one of the actions:

1. Replace the effect variable $Y$ with an attribute that would not be an effect variable in any of the stories. Such replacement variables include: "lip thickness", "earthquakes", "lactose intolerance", "rainfall", "is allergic to peanuts", "brown eyes", "curly hair", "black hair", "foot size", "freckles"

2. Create an irrelevant treatment variable $X$ that does not play a causal role in any of our commonsensical stories. Such as: "can swim", "is religious", "has a brother", "has visited England", "likes spicy food", "is vegetarian", "speaks english", "drinks coffee", "plays card games", "listens to jazz", "solar eclipse", "has a sister", "full moon"

To transform a commonsensical story into an anti-commonsensical story, we apply one of these replacements sampled uniformly, resulting in stories such as:

- Ability to swim has a direct effect on studying habit and exam score. Studying habit has a direct effect on exam score.
- Gender has a direct effect on department competitiveness and peanut allergy. Department competitiveness has a direct effect on peanut allergy.

- Liking spicy food has a direct effect on relationship status. Appearance has a direct effect on relationship status.
- Playing card games has a direct effect on diabetes and lifespan. Smoking has a direct effect on diabetes and lifespan. Diabetes has a direct effect on lifespan. Smoking is unobserved.

For a full list of the replacements and how the replacements are made, check out the code.

## A.8 Explanation Template

Step ① Extract the causal graph: The causal graph expressed in the context is: "$\mathcal{G}$".

Step ② Identify the query type: The query type of the above question is "query_type".

Step ③ Formulate the query to its symbolic form: The formal form of the query is "symbolic_expression".

Step ④ Collect all the available data: The available data are: "$\boldsymbol{d}$".

Step ⑤ Derive the estimand: Based on the graph structure and causal query, the question can be simplified into estimand "est".

Step ⑥ Solve for the estimand: Plug in the available data "$\boldsymbol{d}$" into "est".
$\text{est}(\boldsymbol{d})$
$\approx \text{float}(a)$

Since the estimate for the estimand is $\text{float}(a)$, the overall answer to the question is $\text{bool}(a)$.

# B  Experimental Details

## B.1  CAUSALCOT Prompt

Q: [question from the dataset]

Guidance: Address the question by following the steps below:

Step 1) Extract the causal graph: Identify the causal graph that depicts the relationships in the scenario. The diagram should simply consist of edges denoted in "var1 -> var2" format, separated by commas.

Step 2) Determine the query type: Identify the type of query implied by the main question. Choices include "marginal probability", "conditional probability", "explaining away effect", "backdoor adjustment set", "average treatment effect", "collider bias", "normal counterfactual question", "average treatment effect on treated", "natural direct effect" or "natural indirect effect". Your answer should only be a term from the list above, enclosed in quotation marks.

Step 3) Formalize the query: Translate the query into its formal mathematical expression based on its type, utilizing the "do(·)" notation or counterfactual notations as needed.

Step 4) Gather all relevant data: Extract all the available data. Your answer should contain nothing but marginal probabilities and conditional probabilities in the form "P(...)=..." or "P(...|...)=...", each probability being separated by a semicolon. Stick to the previously mentioned denotations for the variables.

Step 5) Deduce the estimand using causal inference: Given all the information above, deduce the estimand using skills such as do-calculus, counterfactual prediction, and the basics of probabilities. Answer step by step.

Step 6) Calculate the estimand: Insert the relevant data in Step 4 into the estimand, perform basic arithmetic calculations, and derive the final answer. There is an identifiable answer. Answer step by step.

A: [LLM previous response]

Q: Based on all the reasoning above, output one word to answer the initial question with just "Yes" or "No".

A: [LLM final answer]

# C  Additional Technical Background for Preliminaries

## C.1  Graphical Models

We adopt the causal inference framework described in [61]. A causal graph $G := (\boldsymbol{V}, \boldsymbol{E})$ consists of a set of $k$ vertices $\boldsymbol{V} : \{V_1, \ldots, V_k\}$ and directed edges $\boldsymbol{E} := \{e_{ij}\}$, where the existence of each $e_{ij}$ means that there is a direct causation from $V_i$ to $V_j$, also denoted as $V_i \rightarrow V_j$. We also introduce some notations to describe the relative positions among the nodes. Following a standard assumption in causality (but see, e.g., [6]), we will assume that $\mathcal{G}$ is a direct acyclic graph (DAG), where we denote the *parents* of a node $V_i$ as $\mathbf{PA}(V_i) := \{V_j | e_{ij} \in \boldsymbol{E}\}$. We denote *descendants* $\mathbf{DE}(V_i) := \{V_j | V_j \rightarrow \cdots \rightarrow V_i \in \boldsymbol{E}\}$ of a node $V_i$ as all the nodes that have at least one direct path leading to a node. We call a node $V_k$ as a *confounder* (i.e., common cause) of the other two nodes $V_i$ and $V_j$ if $e_{ki}, e_{kj} \in \boldsymbol{E}$; a *collider* (i.e., common effect) if $e_{ik}, e_{jk} \in \boldsymbol{E}$; and a *mediator* if $e_{ik}, e_{kj} \in \boldsymbol{E}$.

Among all the variables in $\boldsymbol{V}$, we use $X$ and $Y$ to denote two special variables, the treatment and effect, respectively.

## C.2  Illustration of the Three Rungs of the Causal Ladder

In Figure 7, we illustrate the difference among the three rungs by enumerating what actions are performed on the variables other than target variables $X$ and $Y$.

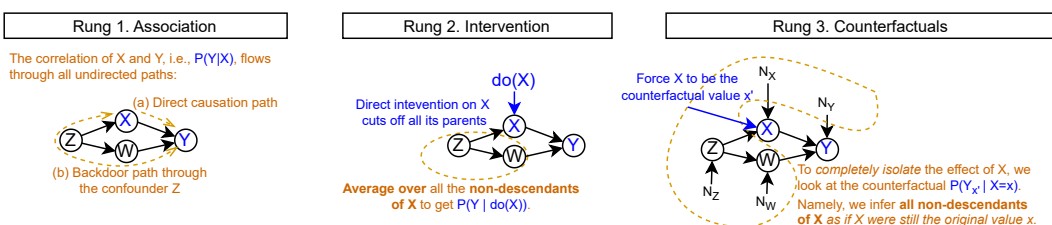

Figure 7: The Causal Ladder consists of three rungs: association, intervention and counterfactuals. We color in blue the treatment $X$ and effect $Y$, as well as the actions on $X$. We color in orange words about how to get the estimand, and we use the orange circle to include all the non-descendants of $X$.

## C.3  Causal Inference Methods

We introduce do-calculus which can downgrade the Rung-2 queries to Rung-1 quantities when it is applicable, and counterfactual predictions which downgrade the Rung-3 queries.

### C.3.1  Do-Calculus

**Do-Operator as a Notation**  As mentioned in Rung 2, the $\mathrm{do}$-operator is a convenient notation to represent an intervention on a variable. For example, $\mathrm{do}(X = x)$ sets the value of variable $X$ to $x$.

**Three Inference Rules for Climbing the Ladder**  Do-calculus is a set of rules that allows us to answer higher-rung questions using lower-rung quantities, such as probability distributions of Rung 1. Given a causal graphical model with and four disjoint sets of variables $X$, $Y$, $Z$, and $W$, and a joint probability distribution that is Markov and faithful to the graph, do-calculus contains the following three rules:

*Rule 1 (Insertion/deletion of observations):*

$$P(Y | \mathrm{do}(X), Z, W) = P(Y | \mathrm{do}(X), W) , \tag{1}$$

if $Y$ and $Z$ are d-separated by $X \cup W$ in $G^*$, the graph obtained from $\mathcal{G}$ by removing all arrows pointing into variables in $X$.

*Rule 2 (Action/observation exchange):*

$$P(Y | \mathrm{do}(X), \mathrm{do}(Z), W) = P(Y | \mathrm{do}(X), Z, W) , \tag{2}$$

if $Y$ and $Z$ are d-separated by $X \cup W$ in $G^\dagger$, the graph obtained from $\mathcal{G}$ by removing all arrows pointing into variables in $X$ and all arrows pointing out of variables in $Z$.

*Rule 3 (Insertion/deletion of actions):*

$$P(Y | \operatorname{do}(X), \operatorname{do}(Z), W) = P(Y | \operatorname{do}(X), W) , \tag{3}$$

if $Y$ and $Z$ are d-separated by $X \cup W$ in $G^\ddagger$, the graph obtained from $\mathcal{G}$ by first removing all arrows pointing into variables in $X$ (thus creating $G^*$) and then removing all arrows pointing into variables in $Z$ that are not ancestors of any variable in $W$ in $G^*$.

These rules are sound and complete [85]. Namely, iff we have all the terms on the right hand side, then the causal term on the left hand side is identifiable.

**Example Application of Do-Calculus** Taking the example in Figure 2, $g_1$ maps the query type ATE to its symbolic expression $\mathbb{E}[Y | \operatorname{do}(X = 1)] - \mathbb{E}[Y | \operatorname{do}(X = 0)]$.

Next, $g_2$ further simplifies the estimand given the confounding graph, as in the flow chart in the middle of Figure 2:

$$\text{ATE} := \mathbb{E}[Y | \operatorname{do}(X = 1)] - \mathbb{E}[Y | \operatorname{do}(X = 0)] \tag{4}$$

$$= \sum_z P(Z = z)[\mathbb{E}(Y|X = 1, Z = z) - \mathbb{E}(Y|X = 0, Z = z)] , \tag{5}$$

which which resolves all the $\operatorname{do}(\cdot)$ terms to probability terms. This example shows the famous backdoor adjustment in do-calculus [59].

### C.3.2 Three Steps for Counterfactual Prediction

Given a SCM $M$, distribution on the exogenous variables $P(u)$, and evidence $e$ from the model $\langle M, P(u) \rangle$, the probability of the counterfactual "if $X$ had been $x$ then $Y$ would have been y, given we observed $e$," denoted $P(Y_x = y|e)$, can be evaluated using the following three steps [67]:

***Abduction:*** Update the probability distribution $P(u)$ by the evidence $e$ to obtain $P(u|e)$

***Action:*** Modify $M$ by the action $do(X = x)$, i.e. replace $X$ with $X = x$ in the structural equations, to obtain the modified SCM $M_x$

***Prediction:*** Use the modified model $\langle M_x, P(u|e) \rangle$, to compute the probability of $Y = y$.

## D Previous Results on CLADDER v1.0

### D.1 Dataset Statistics for v1.0

| | Total | Rung 1 | Rung 2 | Rung 3 |
|---|---|---|---|---|
| **Size** | | | | |
| # Samples | 10,560 | 3,288 | 3,288 | 3,984 |
| **Question** | | | | |
| # Sentences/Sample | 6.85 | 6.00 | 7.00 | 7.25 |
| # Words/Sample | 94.47 | 76.41 | 96.84 | 103.42 |
| # Nodes/Graph | 3.54 | 3.54 | 3.55 | 3.54 |
| # Edges/Graph | 3.44 | 3.41 | 3.43 | 3.46 |
| **Answer** | | | | |
| Positive Class (%) | 50 | 50 | 50 | 50 |
| **Explanations** | | | | |
| # Sentences/Sample | 13.11 | 12.04 | 13.76 | 13.83 |
| # Words/Sample | 146.82 | 141.88 | 147.88 | 151.30 |

Table 5: Statistics of our CLADDER data v1.0.

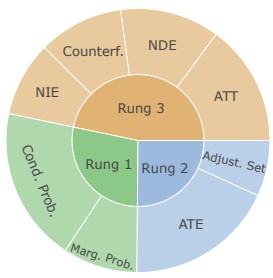

Figure 8: Distributions of query types in our dataset v1.0.

Our data-generating procedure has the potential to algorithmically generate very large amounts of questions. In practice, we pick a dataset size that is large enough to be representative, and at the same time not too large to be problematic given the expensive inference costs of LLMs. We therefore set our dataset size to be 10K. We report the statistics of our dataset in Table 5.

The dataset roughly balanced across the query types, graph structures, stories, and ground-truth answers (as seen in Figure 8). Note that there are some slight adjustments such as more samples for ATE because it allows us to test various techniques, including backdoor and front door adjustments. More details on our design choices can be found in Appendix A.4.

### D.2 Main Results on v1.0

| | Overall Acc. | Acc. by Rung | | | Acc. by Empirical Alignment | | |
|---|---|---|---|---|---|---|---|
| | | 1 | 2 | 3 | Anti-C. | Nonsens. | Comm. |
| Random | 49.27 | 50.28 | 48.40 | 49.12 | 49.69 | 49.01 | 49.12 |
| LLaMa | 45.22 | 63.33 | 31.10 | 41.45 | 45.31 | 45.21 | 45.12 |
| Alpaca | 45.54 | 63.33 | 31.57 | 41.91 | 45.94 | 45.21 | 45.49 |
| GPT-3 Non-Instr. (davinci) | 47.42 | 63.88 | 32.99 | 44.89 | 47.0 | 48.28 | 46.97 |
| GPT-3 Instr. (text-davinci-001) | 57.07 | 63.95 | 63.63 | 48.04 | 59.12 | 57.81 | 54.28 |
| GPT-3 Instr. (text-davinci-002) | 56.24 | 46.03 | 69.55 | 55.04 | 54.75 | 59.65 | 54.31 |
| GPT-3 Instr. (text-davinci-003) | 62.69 | 58.0 | 80.83 | 54.52 | 63.93 | 62.09 | 62.05 |
| GPT-3.5 (queried in May 2023) | 61.71 | 65.12 | 69.9 | 54.11 | 65.43 | 55.15 | 64.55 |
| GPT-4 (queried in May 2023) | 64.28 | 53.94 | 81.87 | 63.11 | 65.75 | 60.87 | 66.21 |
| + CAUSALCoT | 66.64 | 61.67 | 86.13 | 58.23 | 69.32 | 63.02 | 67.60 |

Table 6: Performance of all models on our CLADDER dataset v1.0. We report the overall accuracy (Acc.), and also fine-grained accuracy by rung and by empirical alignment.

We compare the performance of all models in Table 6. First, we can see that the causal reasoning task in CLADDER is in general very challenging for all models. And models such as the earlier, non-instruction-tuned GPT-3 and both LLaMa and Alpaca are no better than random performance. With instruction-tuning, models start to show some improvement. And amongst all, our CAUSALCoT achieves the highest performance of 66.64%, which is 2.36 points better than vanilla GPT-4.

Moreover, from the accuracy by empirical alignment level in Table 6, we can see that the original GPT-4 model performs the best on commonsensical data, but 5.34 points worse on nonsensical data. However, our CAUSALCoT enhances the reasoning ability across all levels, with substantial improvement on anti-commonsensical data and nonsensical data, indicating that CAUSALCoT is particularly beneficial on unseen data.

### D.3 Ablation Study on v1.0

We conduct an ablation study for our multi-step CAUSALCoT. We ablate each of the four subquestions, and observe in Table 7 that classifying the query type and formalizing it has the most effect on the model's performance, which might be because that they are the crucial formalization step in order to do the causal inference correctly. Meanwhile, removing Steps ① and ④, which are mostly about parsing the prompt correctly, have the least impact on performance.

| | Acc. |
|---|---|
| CAUSALCoT | 66.64 |
| w/o Step ① | 64.54 |
| w/o Step ② | 63.74 |
| w/o Step ③ | 63.43 |
| w/o Step ④ | 64.47 |

Table 7: Ablation study.

## E More Experiments

### E.1 Details of Our Error Analysis

For Step 2 about the query type prediction, we report the overall F1 classification score, and also F1 by rungs. For the rest of the steps, we manually annotate the correctness of 100 samples of CAUSALCoT. We report the correctness of est by accuracy, and the correctness of the predicted set of available data by taking the F1 with the ground-truth $d$. For Step 5, we report the accuracy of whether the model simplifies the estimand correctly to est$'$ using causal inference, and also arithmetic correctness (Arith.).

### E.2 ROSCOE Evaluation

We employed the ROSCOE suite of evaluation metrics on step-by-step text reasoning, as introduced by [25], to automate the evaluation of the outputs from CAUSALCoT on 2,000 randomly sampled questions from our dataset. Differing from conventional metrics, ROSCOE is specifically designed to scrutinize the quality of large language model outputs, focusing on aspects such as semantic consistency, logicality, informativeness, fluency, and factuality, all evaluated within the context of step-by-step reasoning, rather than solely the final response. This allows for a more objective and comprehensive assessment of a model's output, greatly aiding in the verification of its interpretability. The results of this evaluation can be found in Table 8 and Figure 9. We consider the model's performance as unsatisfying if it falls out of the top quantile, namely receiving a score $s \in [0, 1]$ smaller than 0.25 when the score should be minimized, or greater than 0.75 when it should be maximized.

We can see in the plot that the good-performing aspects are faithfulness to the original question, reasoning alignment with the ground truth, and absence of external hallucinations, which are consistently within the top quantile. This suggests that the model carries out accurate reasoning within the constraints of the fictitious world introduced in each question.

However, there are some performance dips in redundancy, perplexity chain, and missing step metrics. The first two could potentially be attributed to complex elements such as graph notation, while the relatively lower "missing step" score warrants further investigation. Despite these observations, this analysis largely aligns with our qualitative understanding of the models' good response ability in answering causal questions in our dataset.

|  | Mean | Std | Min | 25% | 50% | 75% | Max |
|---|---|---|---|---|---|---|---|
| Faithfulness | 0.89 | 0.02 | 0.83 | 0.88 | 0.89 | 0.90 | 0.93 |
| Informativeness Step | 0.88 | 0.01 | 0.83 | 0.87 | 0.88 | 0.89 | 0.92 |
| Informativeness Chain | 0.88 | 0.03 | 0.76 | 0.87 | 0.89 | 0.90 | 0.96 |
| Faithfulness Word | 0.95 | 0.01 | 0.92 | 0.94 | 0.95 | 0.96 | 0.97 |
| Repetition Word | 0.02 | 0.02 | -0.00 | 0.00 | 0.02 | 0.04 | 0.05 |
| Repetition Step | 0.02 | 0.01 | -0.00 | 0.00 | 0.01 | 0.03 | 0.06 |
| Reasoning Alignment | 0.92 | 0.01 | 0.86 | 0.91 | 0.92 | 0.93 | 0.95 |
| External Hallucination | 0.97 | 0.02 | 0.84 | 0.96 | 0.97 | 0.98 | 0.99 |
| Redundancy | 0.80 | 0.05 | 0.56 | 0.77 | 0.80 | 0.83 | 0.92 |
| Common Sense Error | 0.95 | 0.01 | 0.86 | 0.94 | 0.95 | 0.96 | 0.98 |
| Missing Step | 0.78 | 0.03 | 0.58 | 0.76 | 0.78 | 0.80 | 0.88 |
| Semantic Coverage Step | 0.99 | 0.01 | 0.95 | 0.98 | 0.99 | 0.99 | 1.00 |
| Semantic Coverage Chain | 0.98 | 0.01 | 0.93 | 0.98 | 0.98 | 0.99 | 0.99 |
| Discourse Representation | 0.06 | 0.13 | 0.00 | 0.01 | 0.01 | 0.05 | 0.67 |
| Coherence Step Vs Step | 0.14 | 0.27 | 0.00 | 0.00 | 0.01 | 0.07 | 0.94 |
| Perplexity Step | 0.02 | 0.01 | 0.00 | 0.02 | 0.02 | 0.03 | 0.07 |
| Perplexity Chain | 0.17 | 0.07 | 0.05 | 0.11 | 0.17 | 0.23 | 0.42 |
| Perplexity Step Max | 0.00 | 0.00 | 0.00 | 0.00 | 0.00 | 0.01 | 0.02 |
| Grammar Step | 0.93 | 0.04 | 0.77 | 0.90 | 0.93 | 0.96 | 0.99 |
| Grammar Step Max | 0.53 | 0.35 | 0.02 | 0.12 | 0.65 | 0.85 | 0.99 |

Table 8: Statistics of ROSCOE scores evaluated on answers from CAUSALCOT on 2,000 randomly sampled questions from our dataset.

## F    Comparison with Existing Causality-Related Datasets

We show in Table 9 the distinction of our work from all existing causality-related datasets that address either the causality-as-knowledge task, or the causality-as-language-comprehension task.

| | Question Types | | | Skill Types | | | |
|---|---|---|---|---|---|---|---|
| | Assoc. | Interv. | Counterf. | CI Method | Formalization of Causal Queries | Causal RE | Qualitative Reasoning |
| ***Datasets for Causality as Knowledge (Commonsense Causality)*** | | | | | | | |
| COPA [2012] | ✗ | ✓ | ✗ | ✗ | ✗ | ✗ | ✗ |
| Event2Mind [2018] | ✗ | ✓ | ✗ | ✗ | ✗ | ✗ | ✗ |
| ATOMIC [2019] | ✗ | ✓ | ✗ | ✗ | ✗ | ✗ | ✗ |
| SocialIQA [2019] | ✗ | ✓ | ✗ | ✗ | ✗ | ✗ | ✗ |
| TimeTravel [2019] | ✗ | ✓ | ✗ | ✗ | ✗ | ✗ | ✗ |
| Goal-Step [2020] | ✗ | ✓ | ✗ | ✗ | ✗ | ✗ | ✗ |
| Abductive (ART) [2020] | ✗ | ✓ | ✗ | ✗ | ✗ | ✗ | ✗ |
| Com2Sense [2021] | ✗ | ✓ | ✗ | ✗ | ✗ | ✗ | ✗ |
| CRASS [2022] | ✗ | ✗ | ✓ | ✗ | ✗ | ✗ | ✗ |
| ***Datasets for Causality as Language Comprehension (Causal Relation Extraction)*** | | | | | | | |
| SemEval2021 Task8 [2010] | ✗ | ✗ | ✗ | ✗ | ✗ | ✓ | ✗ |
| EventCausality [2011] | ✗ | ✗ | ✗ | ✗ | ✗ | ✓ | ✗ |
| Causal-TimeBank [2014] | ✗ | ✗ | ✗ | ✗ | ✗ | ✓ | ✗ |
| CaTeRS [2016] | ✗ | ✗ | ✗ | ✗ | ✗ | ✓ | ✗ |
| BECauSE [2017] | ✗ | ✗ | ✗ | ✗ | ✗ | ✓ | ✗ |
| TellMeWhy [2021] | ✗ | ✗ | ✗ | ✗ | ✗ | ✓ | ✗ |
| ***Datasets for Formal Causal Reasoning*** | | | | | | | |
| Corr2Cause [42] | ✗ | ✓ | ✗ | ✓ | ✓ | ✗ | ✗ |
| CLADDER (Ours) | ✓ | ✓ | ✓ | ✓ | ✓ | ✓ | ✓ |

Table 9: Comparison of our dataset and existing causal or reasoning datasets. The aim of our dataset is to test the pure reasoning ability of LLMs on causal questions. For each dataset, we first identify whether its question types cover the three rungs: association (Assoc.), intervention (Interv.), and counterfactuals (Counterf.). We also check what skill types the dataset tests: the application of causal inference methods (CI Method), formalization of causal queries, causal relation extraction from the given text (Causal RE), and qualitative reasoning.

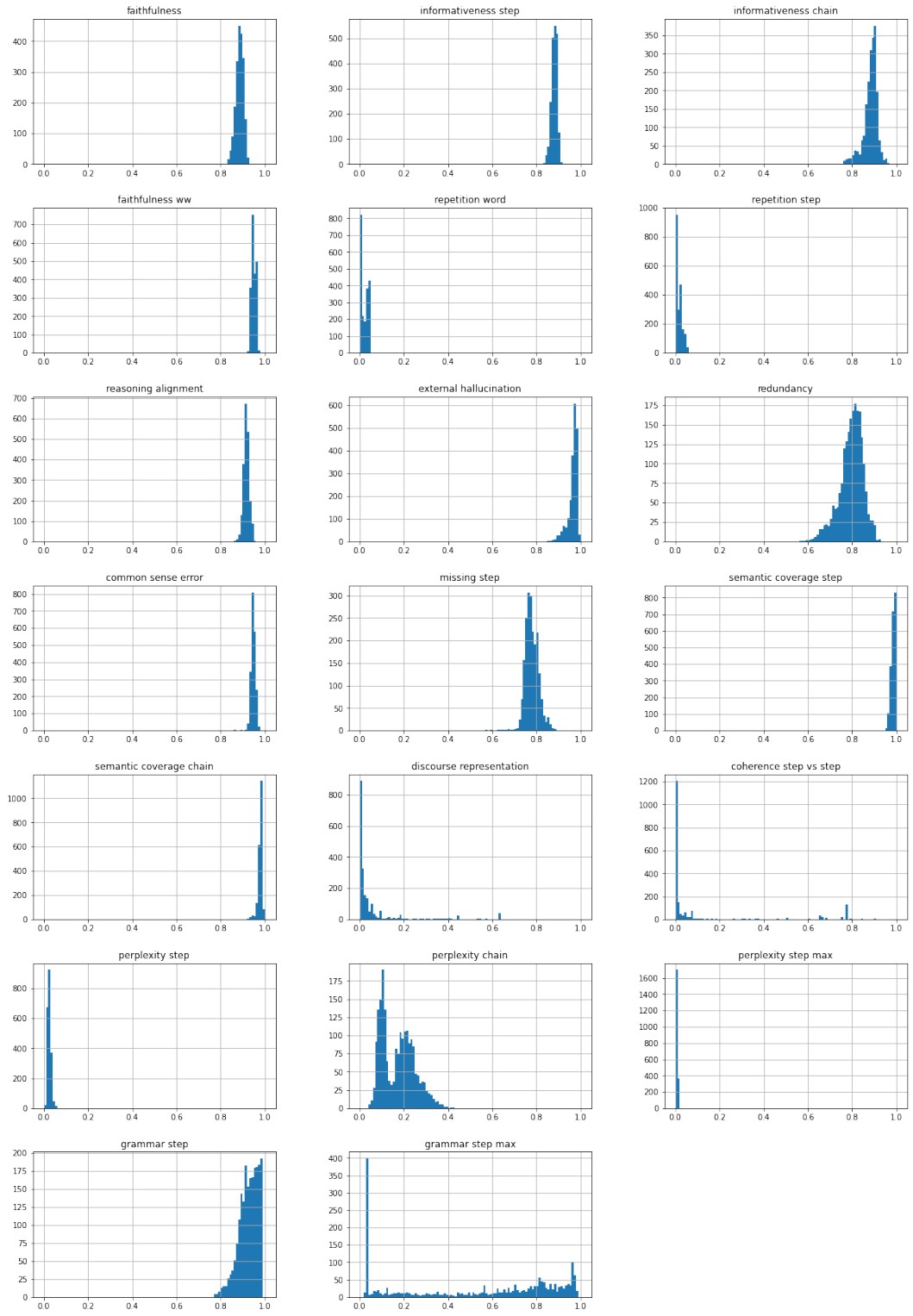

Figure 9: ROSCOE scores of answers from CAUSALCOT on 2,000 randomly sampled questions from our dataset.

