# OpenReview forum: "CLadder: Assessing Causal Reasoning in Language Models"
_NeurIPS.cc/2023/Conference — NeurIPS 2023 poster_

### Official Review · Reviewer_YWrR · 2023-07-04

**Soundness:** 2 fair
**Presentation:** 3 good
**Contribution:** 3 good
**Rating:** 6
**Confidence:** 4

**Summary:**

The paper collects a new dataset, named Cladder, that aims at evaluating the causal reasoning abilities of language models. The datasets contain 10,000 questions asking about boolean questions that require intensive causal reasoning.
The dataset is constructed based on the idea of a causal inference engine, which covers three rungs of causal inference. Specifically, the authors first sample formal representations of causal inference problems by sampling causal graphs as well as query types. Next, the formal representations are verbalized into natural language problems, where the variables in the formal representations are named with concepts in the stories collected from commonly cited causal inference books. Finally, the representations are verbalized into natural language problems using fixed templates.

Using the collected datasets, the paper evaluates a line of language models’ causal reasoning abilities. Experiments results suggest that these problems are still hard for language models in the zero-shot setting (if I understand correctly). In addition, the paper proposes CausalCoT which lets LMs explicitly state causal problems and then solve the problem. The results suggest CausalCoT is better than the baseline.


**Strengths:**

The paper works on an interesting problem, causal inference, and collects a useful dataset consisting of 1,0000 examples.

The data creation process is well-designed. The dataset covers multiple rungs and multiple causal graphs. In addition, the alignment also includes anti-commonsense and nonsensical settings, which is essential for controlling possible effects of memorization in LLMs.

The paper includes additional error analysis highlighting the LM’s capabilities in performing different steps in the process of solving causal inference problems.

The paper is well-written and easy to follow.


**Weaknesses:**

1: The tested baselines (in Table 2) need clarification. Also, the baselines might be somewhat weak.

Are the methods tested in Table 2 implemented in zero-shot setting? (I am guessing that based on section 5.6). I wouldn’t be surprised if these language models utterly fail in a zero-shot setting.

While the paper tries to frame that providing in-context examples is orthogonal to the paper (section 5.6). I believe it is necessary to benchmark what much better systems can do (e.g., few-shot CoT prompting or few-shot CausalCoT prompting) to understand the capabilities of LMs as well as the potential headroom.

2: The language is synthetic.

As suggested in 3.2, the verbalization of the formal causal problem is rule-based, which could result in synthetic and formulaic language. The paper does not provide much evaluation of the language side of the datasets in the main body.

3: No human performance

The paper does not provide human performance for this dataset. Although many problems are adapted from books, the templated-based verbalization as well as the nonsensical setting could make some problems tricky even for humans. Thus, human evaluation is still valuable. Also, it can help verify the quality of the datasets.


**Questions:**

Are the baselines all in zero-shot setting?

In addition, it would be helpful if the authors can include some full prompt examples in the appendix.



**Limitations:**

The paper discusses the limitations in Section 7.

---

> ### Author Rebuttal · Authors · 2023-08-10
>
> We thank the reviewer for the overall positive comments about the meaningfulness of the problem (“The paper works on an interesting problem”), dataset contribution (“a useful dataset consisting of 1,0000 examples”), “well-designed” “data creation process” that “covers multiple rungs and multiple causal graphs” and “includes anti-commonsense and nonsensical settings”, comprehensive experiments with “additional error analysis”, and “well-written and easy to follow” writing.
>
> In the following, we address the three major comments raised by the reviewer.
>
> ### 1. Additional Few-Shot Experiments
>
> > Are the baselines all in zero-shot setting?
>
> Yes, our main results (Section 5.2) are zero-shot, and our Section 5.6 analyzes the few-shot/in-context learning. Following your suggestions, we also dedicated a few-shot experiment and reported the performance below:
>
> ||0-Shot|Few-Shot (10-Shot)|
> |----|---|---|
> |GPT-3 Non-Instr. (davinci)|47.42|49.83|
> |GPT-3 Instr. (davinci-001)|57.07|57.19|
> |GPT-3 Instr. (davinci-002)|56.24|57.36|
> |GPT-3 Instr. (davinci-003)|62.69|63.50|
> |GPT-3.5|61.71|61.85|
> |GPT-4|64.28|65.43|
> |+CausalCoT|66.64|70.09|
>
> Several key takeaways are (1) few shot brings some improvement, but relatively minor for most models, maybe because the task is really complex. (2) The strongest model, few-shot CausalCoT, reaches 70.09%. The 4 point improvement could be because CausalCoT uses much richer information to complete the task, thus benefiting more learning from examples.
>
> > I believe it is necessary to benchmark what much better systems can do (e.g., few-shot CausalCoT) to understand the capabilities of LMs as well as the potential headroom.
>
> Thank you for your advice. We implemented the suggestion, and find that few-shot CausalCoT reaches 70.09%, +4 point increase over the 0-shot CoT. We hope this result helps to better understand the capabilities of LMs as well as the potential headroom.
>
> ### 2. Evaluation of the Natural Language
>
> > The paper does not provide much evaluation of the language side of the datasets
>
> In the composition of this work, we followed the standards of dataset creation and experiments in previous work on formal tasks for LLMs, such as this recent [ICLR 2023 paper](https://openreview.net/forum?id=qFVVBzXxR2V) on formal reasoning in logic. Similarly, their language is also formulaic. Nonetheless, we are happy to add evaluations of the quality of the natural language in our data below.
>
> **Evaluation:** For our benchmark dataset, we think the set of criteria to judge it is: (1) grammaticality, (2) human readability, (3) formal correctness, and (4) naturalness, whose performance we report below.
>
> 1. **Grammaticality:** We run the grammar check on our dataset by the LanguageTool package, and get on average 1.26 grammatical errors per 100 words (i.e., 98.74% correct), which shows that most of our language follows English grammar.
> 2. **Human readability** (i.e., how comprehensible/intelligible are the questions to, e.g., a student who has taken a causality course): 96%. We obtain this score by sampling 50 random questions, and let a grad student annotator go through the questions to mark the questions that they cannot understand.
> 3. **Formal correctness:** 100%, because of our systematic generation process together with the formal solutions.
> 4. **Naturalness/Perplexity:** In addition, we also produce how natural the questions sound by the automatic metric of perplexity (since human evaluation might be subjective). We use gpt2 to calculate the perplexity scores, and obtained a score of 21.17 (the lower, the more natural the language sounds to GPT). For comparison, the perplexity of the MATH Dataset (NeurIPS 2021, by [Hendrycks et al.](https://arxiv.org/abs/2103.03874)) is 57.45, which means that the language in our data is more natural-sounding.
>
> For a more intuitive feeling, please feel free to check the examples of our dataset in Appendix Tables 5-12.
>
> **Camera Ready:** We are happy to add the above evaluation scores to the camera-ready version of our dataset.
>
> ### 3. Reporting the Human Performance
>
> > Some problems could be tricky even for humans. Thus, human evaluation is still valuable.
>
> We agree with the reviewer’s intuition that the problems can be challenging for humans too. In addition to the readability score, we also conducted a small experiment getting our coauthors to work through 50 random samples. In these preliminary results, we obtained a correctness score of around 80% as the human performance score. We plan to extend this to a larger cohort for the camera-ready version.
>
> **Camera Ready Plan:** We plan to use the human evaluation score in the following way:
> - The score indicates the human performance, against which we can illustrate whether LLMs surpass it or not.
> - We want to emphasize that the human performance will only be used as a baseline, but not a judgment for the dataset quality. Imagine a university exam that is valid and good-quality, but challenging for the students. Hence, **a good-quality dataset** (i.e., formally correct and readable) **can also be challenging for humans**, and requiring specific skills to solve. That’s another motivation for using models to automate these challenging tasks for humans.
> - Another disclaimer is that, similar to any cognitive science studies, the human performance will be a function of the subjects’ background, including their previous education background, familiarity with causal formulations, and carefulness when solving these problems. Our reported performance is based on a cohort of researchers familiar with the study and took the questions with caution.
>
> > it would be helpful to include some full prompt examples in the appendix.
>
> Sure, we can provide the full prompts in our camera ready version. We composed several examples of our prompts at this anonymous link: https://anonymous-link.notion.site/CLadder-031085fc56854955bcf3d30d499f0f42

---

> > ### Comment · Reviewer_YWrR · 2023-08-11
> >
> > Thanks for your clarification. I appreciate the added experiments and human evaluation, which helps address my concern. I am happy to raise my score.

---

> > > ### Author Response · Authors · 2023-08-11
> > >
> > > Thank you for reading our rebuttal materials. We are glad that the newly added experiments and human evaluations help clarify your concerns. And we appreciate your raise of score and positive support. Have a good weekend :)!

---

### Official Review · Reviewer_rbbm · 2023-07-05

**Soundness:** 3 good
**Presentation:** 4 excellent
**Contribution:** 3 good
**Rating:** 7
**Confidence:** 4

**Summary:**

The paper introduces a large dataset, CLADDER, designed to evaluate the ability of LLMs to perform causal inference. The dataset includes 10K binary questions about variable association and treatment efficacy along with a ground truth explanation. Questions are answered in the context of an hypothetical closed world in which the available measurements allow to unambiguously answer the questions.

Most of the classical causality related NLP tasks focus on commonsense reasoning and domain knowledge, assessing the ability of the model to make deductions based on knowledge of properties of an object or human behavior in everyday life situations. The CLADDER data goes beyond domain knowledge, requiring the model to perform causal inference: inferring the causal graph, describing the treatment effect of interest related to the question (the estimand), choosing an estimator and expressing the numerical results.

To build the dataset, a causal graph derived from a list of 10 classical structures is sampled. A query type, describing the treatment effect of interest is then derived. The underlying estimand is then derived and an estimate is obtained using the rules of do-calculus and the abduction-action-prediction steps of counterfactual prediction. All those components, causal graph, query and results are then translated into natural language to be ingested by classical models. For each causal graph a set of 2 to 5 stories taken from the literature are used to ensure that the variable names and their relation are not nonsensical. A template is also used to generate step-by-step explanations.

Eight models are then evaluated on the task, with an overall accuracy of 66.64 being reached for GPT-4 and CausalCot, a chain-of-thought prompting strategy that relies on 4 sub questions, corresponding to natural reasoning steps, to guide the model. A nonsensical evaluation, accounting for potential data contamination is also performed. The evaluation is further refined for the CausalCot model, by including an ablation study and assessing the effect of in context learning.

**Strengths:**

Classical causality related NLP tasks mostly focused on commonsense reasoning and domain knowledge. The paper goes beyond that, introducing a new original task, CLADDER, which theoretically requires the model to infer the causal graph, describe the treatment effect of interest, choose an estimator and provide an estimate. The paper nicely builds on the framework proposed by Pearl with a dataset of 10K questions covering a large variety of causal graph structures and estimands. Additionally, the explanation generated together with the questions allows further assessment of the reasoning ability of the model.

The evaluation of state of the art models on the task is well conducted with a particular methodology to isolate the effect of data contamination. Additionally, a methodology to boost model's performance using chain-of-thought prompting is proposed. If the improvement is marginal, it allows further investigation on the ability of the model to perform the natural reasoning steps to complete the task.

**Weaknesses:**

To better understand how the information provided is used, it could be interesting to mention a new attribute in the text, completely unrelated for which you provide the marginal probability. Given the low performance, adding noise to the available information is not the priority but it could be nice for future work.

On top of the ablation study, it would be interesting to know how the CausalCoT model performs if it was given the question for step 1 along with the answer and then only the question for step 2 to 4. Then doing the same but providing the answer for step 1 and 2. And finally the same for step 1, 2 and 3. It would provide insights on the ability of the model to perform the task if it received partial answers.

**Questions:**

Line 156 _"processes we define specify all necessary information"_  A bit heavy with two verbs following one after the other. I suggest rephrasing, maybe as: "the causal processes defined encapsulates all necessary..."

In Figure 4, _Translate the question to a formal estimand_, please note that the notation used with a _\sum_ can be ambiguous as this is only valid for an infinite amount of data (ambiguous outside of the Pearl's community). The estimand doesn't depend on the data observed, the estimator does.

**Limitations:**

The conclusions made are only valid in a self contained, hypothetical world without any unmentioned factor or causal relationship. A potential negative impact would be for the model to wrongly infer causal effect in the presence of many unmeasured confounders where conditional and marginal are available for few variables. If widely deployed, that kind of model should be able to state the hypothesis it is relying on or state that not enough information is available to draw conclusions on efficacy.

---

> ### Author Rebuttal · Authors · 2023-08-10
>
> We thank the reviewer for providing very detailed comments and an elaborate summary of the work. We appreciate the reviewer’s comments that “The paper nicely builds on the framework proposed by Pearl with a dataset of 10K questions covering a large variety of causal graph structures and estimands.” and “The evaluation of state of the art models on the task is well conducted with a particular methodology to isolate the effect of data contamination.”
>
> Here we address some remaining questions and comments of the reviewer.
>
> > To better understand how the information provided is used, it could be interesting to mention a new attribute in the text, completely unrelated for which you provide the marginal probability. Given the low performance, adding noise to the available information is not the priority but it could be nice for future work.
>
> This is a perfect suggestion! Actually, we are working on a follow-up paper using this idea to see what perturbation in the prompt will distract the model. And we agree, these explorations (in lines of adversarial attack, or LLM performance analysis) are very interesting to work on.
>
> > On top of the ablation study, it would be interesting to know how the CausalCoT model performs if it was given the question for step 1 along with the answer and then only the question for step 2 to 4. Then doing the same but providing the answer for step 1 and 2. And finally the same for step 1, 2 and 3. It would provide insights on the ability of the model to perform the task if it received partial answers.
>
> This is a nice thought. We conduct the aforementioned experiments, and report the results in the following:
> | Providing the answer of | Asking the question of | Performance |
> | ----------------------- | ---------------------- | ----------- |
> | None                    | Step 1, 2, 3, 4        | 66.64       |
> | Step 1                  | Step 2, 3, 4           | 68.10       |
> | Step 1, 2               | Step 3, 4              | 71.03       |
> | Step 1, 2, 3            | Step 4                 | 72.49       |
> | Step 1, 2, 3, 4         | None (Just final Q)    | 74.51       |
>
> There is a very nice monotonically increasing trend as we provide more partial answers in the CausalCoT process. We are happy to put this result table also in the camera ready version of the paper.
>
>
> > In Figure 4, Translate the question to a formal estimand, please note that the notation used with a \sum can be ambiguous as this is only valid for an infinite amount of data (ambiguous outside of the Pearl's community).
>
> In our work, the word “data” is used to refer to population-level information, as opposed to finite samples: this is indeed unusual, although consistent with how it is often used by Judea Pearl et al. to stress that certain key problems of causal inference are orthogonal to estimation from finite samples. We will add a footnote and a clarification in the image caption to stress this.
>
> > Line 156 "processes we define specify all necessary information" A bit heavy with two verbs following one after the other. I suggest rephrasing, maybe as: "the causal processes defined encapsulates all necessary..."
>
> Thank you for the suggestion. We will improve the presentation of the camera-ready version of our paper accordingly. Thank you for the two notes!

---

> > ### Comment · Reviewer_rbbm · 2023-08-14
> >
> > Thanks a lot for adding this analysis to the ablation study. It's reassuring to observe a monotonically increasing trend when adding the different steps. I really appreciate your answers and the quality of the paper. I recommend it for acceptance.

---

> > > ### Author Response · Authors · 2023-08-14
> > >
> > > Thank you very much for taking time to go through our rebuttal materials. We are glad that the ablation study result provide more insights into CausalCoT, and thank you for suggesting it at the first place. Also, we appreciate your endorsement for the paper, which supports us to continue exploring on this line of research.

---

### Official Review · Reviewer_hcc4 · 2023-07-05

**Soundness:** 2 fair
**Presentation:** 3 good
**Contribution:** 3 good
**Rating:** 5
**Confidence:** 3

**Summary:**

The author proposes a math-word-style formal causal reasoning dataset, CLADDER, which is the first one that can be used for formally evaluating the causal reasoning/inference ability of LLMs. The authors also proposed a new prompting strategy, CausalCoT, which improves the zero-shot performance of GPT-4 by 2.4% on the proposed dataset.

**Strengths:**

1. The paper is well-written and easy to follow. Illustrations are helpful.

2. It is the first formal causal inference/reasoning dataset in a natural language, math-word-problem-like format, which provide a nice addition to the current logical/math reasoning datasets.

3. It provides a way to formally evaluate the causal reasoning ability of LLMs for the first time.

**Weaknesses:**

1. The causal inference questions used in the dataset seem to be a bit simple for evaluating GP3/GPT4 scale LLMs. To my understanding, they only require a one-step application of a causal inference equation to binary variables. It would be great if hard questions with multi-step reasoning could be included.

2. To my understanding, only zero-shot prompting is used to evaluate the performance of LLMs in Table 2, while the common evaluation paradigm for the reasoning abilities of LLMs is through few-shot CoT prompting. It would be helpful if the authors provided another set of results with few-shot prompting.

3. It would be helpful to actually fine-tune a small LLM (e.g., 7B LLaMa) on the proposed dataset since the dataset is large enough to be trained on (10K). In this case, we can have a better understanding of the difficulty of the proposed dataset, and whether there are spurious correlations/shortcut features in the dataset that can be captured by a statistical model.

4. The authors only apply the proposed CausalCoT method to GPT4, and the improvement seems to be small (2.4%). The evaluation of the proposed prompting technique seems to be insufficient. I'm wondering what is the results when CausalCoT is applied to other models and when it is applied with few-shot in-context learning.

**Questions:**

Please see weaknesses.

**Limitations:**

Yes

---

> ### Author Rebuttal · Authors · 2023-08-10
>
> We thank the reviewer for their positive feedback that “The paper is well-written and easy to follow”, the dataset is valuable as “It is the first formal causal inference/reasoning dataset in a natural language, math-word-problem-like format”, which constitutes “a nice addition to the current logical/math reasoning datasets”, and “It provides a way to formally evaluate the causal reasoning ability of LLMs for the first time.”
>
>
> Below we address several comments.
>
> > To my understanding, they only require a one-step application of a causal inference equation to binary variables. It would be great if hard questions with multi-step reasoning could be included.
>
> This comment might stem from the misunderstanding that our questions require one-step reasoning: we explain below that this is not the case, and we will clarify this in a revised version of our paper.
>
> As shown in Figure 1, correctly solving our causal questions needs five steps. The last step (Step 5) of the solution corresponds to the idea “a one-step application of a causal inference equation[s]”, which might have led to the confusion. For the remaining four steps, they involve different subskills such as (1) parsing the causal graph by causal relation extraction, (2) classifying the query type, (3) deriving the estimand by causal inference (e.g., through backdoor adjustment), and (4) collecting available data by semantic parsing. In particular, to generate ground truth solutions, step (3) is taken care of by the causal inference engine, which in general requires non-trivial reasoning about an underlying causal graph and additional assumptions.
>
> This 5-step reasoning process makes our CLadder dataset actually very challenging for the current LLMs, with GPT4 reaching only 64.28% accuracy.
>
> As for the connection with other multi-step reasoning works such as HotpotQA, it can be an interesting extension to do causal inference several times to get the final answer, or do a formal causal inference + commonsense reasoning combination, by letting the LLM to use common knowledge (e.g., age affects experience but not vice versa) to build the causal graph.
>
> > It would be helpful if the authors provided another set of results with few-shot prompting.
>
> Our Section 5.6 has a small-scale experiment to analyze the effect of few-shot/in-context learning. Following your suggestions, we also dedicated a few-shot experiment and reported the performance below:
>
> ||0-Shot|Few-Shot (10-Shot)|
> |----|---|---|
> |GPT-3 Non-Instr. (davinci)|47.42|49.83|
> |GPT-3 Instr. (davinci-001)|57.07|57.19|
> |GPT-3 Instr. (davinci-002)|56.24|57.36|
> |GPT-3 Instr. (davinci-003)|62.69|63.50|
> |GPT-3.5|61.71|61.85|
> |GPT-4|64.28|65.43|
> |+ CausalCoT|66.64|70.09|
>
> Several key takeaways are (1) few shot brings some improvement, but relatively minor for most models, maybe because the task is really complex. (2) The strongest model, few-shot CausalCoT, reaches 70.09%. The +4 point improvement could be because CausalCoT uses much richer information to complete the task, thus benefiting more learning from examples.
>
> > It would be helpful to actually fine-tune a small LLM (e.g., 7B LLaMa) on the proposed dataset since the dataset is large enough to be trained on (10K). In this case, we can have a better understanding of the difficulty of the proposed dataset, and whether there are spurious correlations/shortcut features in the dataset that can be captured by a statistical model.
>
> We carefully designed the dataset to avoid trivial shortcut features by, for example, balancing the number of times the correct answer is “yes” vs “no” across all stories and query types. Additionally, to minimize any correlation between for every question we generate both “polarities” (such that the same question can be asked with the correct answer being “yes” or “no”). For example, for a question like “Is ringing alarm less likely than silent alarm overall?” we also generate the positive polarity version “Is ringing alarm more likely than silent alarm overall?” and then randomly select one of the verbalizations with the corresponding answer.
>
> Furthermore, note that our benchmark is a challenge set rather than a train+test set, much like the other challenge sets TruthfulQA (ACL 2022), and MoralExceptQA (NeurIPS 2022). For challenge sets such spurious correlations are generally less of a concern, because no training is involved to let the model learn/overfit to any undesirable “shortcuts.”

---

> > ### Author Response · Authors · 2023-08-17
> >
> > Dear reviewer, we have read your review carefully, conducted the requested experiment, and included a detailed reply in the rebuttal. Could you let us know if you have further questions? Happy to follow up :)!

---

> > > ### Comment · Reviewer_hcc4 · 2023-08-19
> > >
> > > Thank you for your rebuttal. I feel relatively positive about the paper as it is the first formal causal inference dataset in NLP, but some concerns still remain, so I decided to keep my original score unchanged. Specifically,
> > >
> > > 1. The evaluation of the proposed prompting technique is not very thorough.
> > >
> > > 2. The dataset is large enough (10K) to fine-tune with open-source LLMs. No one will actually evaluate on a 10K dataset. I think some sort of fine-tuning can reveal potential issues of the proposed dataset. e.g. whether there are undiscovered shortcuts, and whether the actual causal reasoning process of this dataset is too easy that the un-fine-tuned models cannot perform well because they rarely saw this causal inference data in the training set, etc. Since training on the proposed dataset to improve the causal reasoning ability of LLMs is also an interesting usage of this dataset, I think it will be very helpful to do the fine-tuning experiments.

---

> > > > ### Author Response · Authors · 2023-08-21
> > > >
> > > > Thank you for positively supporting our work. We are glad to take further efforts to address your remaining concerns.
> > > >
> > > > **Re the dataset size (10K):**
> > > >
> > > > One advantage of our work is that the number of possible questions can be flexibly downsampled or upsampled, so 10K is not a fixed number. Our major contribution is to provide a systematic way (and we also open-source our code) to generate causal inference questions at finger tip and the users can control many possible combinations of graphs, queries etc. So it is also an option to publish a 1K version of the data for the evaluation mode.
> > > >
> > > > **Re the potential shortcuts:**
> > > >
> > > > We are conducting the finetuning experiments now. It will take some time. In case the results did not get into the rebuttal time window, we can add it in the camera-ready version.
> > > >
> > > > Here are some preliminary results:
> > > > 1. We are training a RoBERTa model now. So far with default parameters, it is still close to random guess with 50% accuracy, and fluctuating a bit between 49% and 51%. We are further finetuning the hyperparameters to see if it achieves higher accuracy.
> > > > 2. In the meantime, we did a search over spurious correlations by calculating how correlated are the ngrams to labels. We report the absolute value of the [normalized point-wise mutual information](https://en.wikipedia.org/wiki/Pointwise_mutual_information#Normalized_pointwise_mutual_information_(npmi)) (nPMI) between each ngram and the label. Note that the value of nPMI is between [-1,+1], where -1 (in the limit) is never occurring together, 0 is independence, and +1 is complete co-occurrence.
> > > >
> > > > - 1-gram (single word): average |nPMI|=0.005
> > > > - 2-gram (two-word combination): average |nPMI|=0.007
> > > > - 3-gram: average |nPMI|=0.011
> > > >
> > > > All of the above |nPMI|s are pretty small, close to 0, which means that they are largely independent, which also echoes with the design of our dataset that balances our data generation process (incl., causal graphs, query types, etc) across the positive and negative labels.
> > > >
> > > > We are happy to add these above results to the future version of the paper to show the quality of our dataset.
> > > >
> > > > **Re “the evaluation of the proposed prompting technique is not very thorough.”:**
> > > >
> > > > Thank you for the note. In the paper, we provide detailed analysis of the prompting technique, which we can highlight some main findings below:
> > > >
> > > > - In Section 5.4 Error Analysis by Subquestions, we can see in Table 3 the performance of each single step. For example, the models deal well with the arithmetics, and causal graph extraction (which are common tasks that LLMs have been more or less trained on), while the Step 2 query type classification is very difficult, which is a crucial contribution of the causal inference community, and a highly technical skill. Our dataset provides a first step to introduce the formal causal inference task to LLMs.
> > > >
> > > > - In Section 5.5, as well as our [additional experiment](https://openreview.net/forum?id=e2wtjx0Yqu&noteId=NLKhXRNjHH) to Reviewer rbbm, we can see the effect of each step on the final accuracy. For example, Step 2 and 3, which are crucial for the formalization of the estimated, contributes the most to the CoT process.
> > > >
> > > > Does the above address your concerns? Feel free to let us know if you need any additional information.

---

### Official Review · Reviewer_5uc1 · 2023-07-14

**Soundness:** 3 good
**Presentation:** 3 good
**Contribution:** 2 fair
**Rating:** 5
**Confidence:** 3

**Summary:**

The research question is whether large language models are capable of doing
formal causal reasoning as formalized by the approach with SCMs popularized by Pearl.
The authors provide a benchmark of 10k examples to evaluate such abilities; they describes in
detail the design choices for this benchmark.
They then propose a chain-of-thought prompting strategy (CausalCoT) to enhance LLMs with causal reasoning abilities; evaluations are carried out both on OpenAI's proprietary LLMs
and open-source LLMs.

**Strengths:**

* The design principles behind the benchmark are well-explained. In particular since LLMs
tend to struggle with arithmetic-heavy tasks authors took care of using binary variables and
simple causal graphs.
* A variety of topologically distinct treatment-effect pairs is considered, covering thus multiple
causal graphs.
* Non-sensical and anti-commonsense variants of stories are injected to reduce the effect of LLMs having
memorized commonsense causal knowledge. Results are reported for the different splits (commonsense, anti-commonsense, etc...).
* Good quality figures and diagrams; sufficient background material in the appendix for those not familiar with Pearl's causal framework.
* A new application of CoT to formal causal inferences.


**Weaknesses:**

* I have some concerns about whether the research question is appropriate for the conference audience. For example, the cited (in the paper) concurrent
work of (Kiciman et al 2023) considers model evaluation on several ways of formalizing causal discovery; this broadens the scope of the potentially interested audience.
* In my opinion it would be very important to discuss why this specific approach to evaluating causality matters
and the ML community should care. For example, one might argue that human beings have deeply advanced their causal understanding
of the world for centuries without using Pearl's framework to draw causal conclusions, so why should we care about LLMs using this specific framework? To draw another analogy,
mathematics keeps progressing, yet almost no professional mathematician can write a formal proof or uses a formal
proof to discover new math; the interest in understanding if LLMs can write formal proofs or can assist in writing formal proofs should be well-motivated.
* If formal causal reasoning abilities matter for discovering new science (e.g.~the quote opening the Introduction),
then it would be good to discuss why a setting in which the model uses software tools for causal inference was not considered. Do such tools exist? For example, for formal mathematical proofs this is the case and a natural approach would be to make LLMs make use of such tools.
* CausalCoT makes Rung 3 worse (Table 2). IIUC Rung 3 is the most challenging task (counterfactuality), so it would be good to explain why CausalCoT makes the accuracy worse.

**Questions:**

* L. 75; is it correct that Rung i+1 is more difficult than Rung i? Maybe it could be emphasized also in this section.
* L. 264: please add an explanation of why temperature=0
* Table 2: Can you explain why CausalCoT makes performance drop on Rung 3, which should be the most challenging one in the Ladder?
* I am having trouble to understand how CausalCoT works in practice. Could you point me to where I can find an interaction with the model
in the supplementary materials? It would be great if one such example could make it to the paper or the appendix. I am sorry if I missed it somewhere.
More specifically, according to this [post](https://www.lesswrong.com/posts/yZb5eFvDoaqB337X5/investigating-causal-understanding-in-llms)
some form of few-shot examples are needed to successfully prompt models, so I am wondering if one needs to do the same to get the causal graph representation when using CausalCoT.

Overall, my main concern is a lack of discussion or argument for why evaluation on this specific formalization of causal reasoning should matter.
I am happy to increase the score if my concerns get addressed.

**Limitations:**

Good. During the rebuttal period the authors agreed to discuss a couple of extra limitations of the work.

---

> ### Author Rebuttal · Authors · 2023-08-10
>
> We thank the reviewer for recognizing that “The design principles behind the benchmark are well-explained”, the dataset is diverse with “A variety of topologically distinct treatment-effect pairs” “covering thus multiple causal graphs”, the experiments are multi-faceted with “Non-sensical and anti-commonsense variants of stories”, and the paper has “Good quality figures and diagrams”.
>
> In the following, we will focus on a few main points:
> 1. The value of the paper, including (a) the relation with the concurrent paper Kiciman et al. (2023), and (b) the value of automating formal reasoning abilities for LLMs/society, and
> 2. Several technical details to clarify.
>
> ### 1. Clarification of the Value of the Paper
>
> > the cited concurrent work of (Kiciman et al 2023) considers model evaluation on several ways of formalizing causal discovery.
>
> Thank you for bringing up the comparison of our paper and this concurrent work (published in May, two weeks before the NeurIPS deadline). Firstly, note that  under the [NeurIPS policy](https://nips.cc/Conferences/2023/PaperInformation/NeurIPS-FAQ#:~:text=Authors%20are%20not%20expected%20to%20compare%20to%20work%20that%20appeared%20only%20a%20month%20or%20two%20before%20the%20deadline.), “Authors are not expected to compare to work that appeared only a month or two before the deadline” including self-published work such as arXiv submissions.
>
> Nonetheless, we are happy to highlight the _orthogonality_ between Kiciman et al. (2023) and our work. Their work falls under the umbrella of “causality as knowledge”, whereas our work focuses on “causality as formal reasoning”, following the categorization in our Abstract and Intro, L4-6 and L33-39.
>
> To illustrate the difference between the two approaches, we use the formulation of “Counterfactual” (Rung 3) as an example:
>
> **Example from our data:**
>
> Q: Effort has a direct effect on college admission. [...] For nonsmokers who are lazy, the probability of college admission is 55%. For nonsmokers who are hard-working, […] If we disregard the mediation effect through effort, would smoking positively affect college admission?
>
> A: Yes. (This answer is obtained by applying the causal inference engine.)
>
> **Example from Kiciman et al (2023)’s data (Table 8):**
>
> Q: A woman does not order Chinese food.
> What would have happened if she had ordered Chinese food?
>
> A: The woman would have become less hungry.
>
> **Difference:** Crucially, our answer deductively follows directly by the rules of formal causal inference and information given in the prompt, whereas a commonsense question relies on a common sense understanding of actions and consequences, which is far more susceptible to implicit biases, e.g., if the woman was ordering for someone else, then her hunger level wouldn’t change at all.
>
> > it would be very important to discuss why this specific approach to evaluating causality matters and the ML community should care.
>
> We share the reviewer’s thoughtfulness concerning the value of our work to the ML community. Here are some of the main aspects to better motivate the value of our contributions:
>
> Our causal inference dataset relies on _formal reasoning_, but not _generating proofs_. Our benchmark occupies a hitherto unaddressed niche of evaluating the formal reasoning abilities of LLMs (much like math word problems or university math exams), but for problems concerning causality.
>
> Now, one might question to what extent it is necessary to evaluate the formal causal reasoning abilities of LLMs. Aside from (1) the academic benefits of understanding the strengths and weaknesses of the current state-of-the-art for LLMs, we believe our work is also a step towards some important practical applications. (2) From analyzing medical studies to risk assessments, formal causal inference problems occur in many high value tasks. Here, as the use of LLMs becomes more common to automate certain tasks, it is important to (3) understand to what extent LLMs can reliably understand and solve such problems. It has been claimed that LLMs understand causality well (e.g., Kiciman et al report very high performance numbers such as 97% and 92%). In contrast, we are the first to show that LLMs have a far longer way to go, reaching only 60+% now on CLadder. It is scientifically worthwhile to study what LLMs can already do and what they cannot.
>
> > it would be good to discuss why a setting in which the model uses software tools for causal inference was not considered. Do such tools exist?
>
> We concur that it could be interesting to connect LLMs to causal inference tools (e.g., TETRAD etc.). However, we argue that one important step towards augmenting LLMs is understanding to what extent they already solve certain reasoning tasks.
>
> ### 2. Technical details
>
> > Table 2: Can you explain why CausalCoT makes performance drop on Rung 3
>
> In short, Rung 3 has difficulty even with the assistance of CausalCoT because it fails severely at one of the steps. In “5.4 Error Analysis by Subquestions”, we can see that in Table 3, one of the trickest steps for LLMs is Step 2, to identify the query type (e.g., ATE, NDE, …), which only reaches 50% F1 in general, and the worst for Rung 3, with only 42% F1.
>
> > why temperature=0?
>
> For reproducibility, otherwise LLMs' performance will fluctuate.
>
> > Could you point me to where I can find an interaction with the CausalCoT model in the supplementary materials?
>
> Thank you for the good suggestion, we will add the example prompts and more experimental details in the Appendix of the improved version of the paper. We will also show here the prompt of our zero-shot and 10-shot experiments.
>
> An example of our **zero-shot prompt** is at this [anonymous link](https://anonymous-link.notion.site/CLadder-031085fc56854955bcf3d30d499f0f42#0df909ae3e064b89afc78817332d6798).
>
> An example of our **10-shot prompt** is at this [anonymous link](https://anonymous-link.notion.site/CLadder-031085fc56854955bcf3d30d499f0f42#bda58d0c708e43c4bb8417bb58b647b8).

---

> > ### Comment · Reviewer_5uc1 · 2023-08-16
> > **Further Questions**
> >
> > Thanks for the explanations and the example prompts. I have some more questions and comments.
> >
> > `Comparison to Kiciman et al.`
> > I am aware of the venue's policy on comparison to concurrent work. While reading the paper under review, the question of why studying this particular formalization of causality is important has come to my mind multiple times. In the related work the authors cite Kiciman et al. and I had a look at that work to get a better understanding of  the current debate around causality and LLMs. I found the way in Kiciman et al. the research topic is *presented* quite compelling. In the paper under review I feel the part from L30-50 would need expansion to make the research question look compelling. For example, understanding the difference between correlation and causation (L36-37) or proposing plausible novel hypotheses  (L38-39) does not really need Pearl's framework. However, one might point out some examples of the necessity or new deep insights / discoveries brought by this causality framework and improve the motivation for the research question. In summary, I do think a bit more work should be put on presenting and motivating the research question.
> >
> > `Table 2: Can you explain why CausalCoT makes performance drop on Rung 3`
> > I am still confused. I understand that there is a drop in Step 2. However, in Table 3 it seems that adding CasualCoT results in about a 5 points drop from GPT-4 on Rung3, which I find surprising as adding the formal casual reasoning with CoT should help on the most challenging Rung 3. Also on these tasks, what magnitude of accuracy difference is statistically significant?
> >
> > `However, we argue that one important step towards augmenting LLMs is understanding to what extent they already solve certain reasoning tasks.`
> > I am worried that if models use the right tools, they might solve these causality tasks easily. For analogy, an LLM can make simple arithmetic mistakes but if it uses a calculator tool in the right way it can get many arithmetic questions right. While I do not think there is a need to add experimental results with tools, I do think this should be pointed out as a potential limitation of this study and its conclusions on the abilities of causal understanding of LLMs.

---

> > > ### Author Response · Authors · 2023-08-17
> > >
> > > `Comparison to Kiciman et al.`
> > > We appreciate your interest in our work. And we appreciate the effort you've taken to delve into the Kiciman et al. work to gain a deeper understanding of our research. However, we believe that we might be on a slippery slope if we start drawing parallels between our work and that work. This might inadvertently contradict the concurrent work policy. So we would prefer for our work to be judged on its own merit.
> > >
> > > Independent of the above, we agree that the difference between correlation and causation is the most obvious and best known aspect of causality; however,Pearl's framework is much richer than this. While Pearl's framework is not the only one dealing with causality, it is probably the most common one in the ML community, so it is a very natural framework to use here.
> > >
> > > For example, another approach to causal inference which may be considered, apart from the one based on structural causal models (SCMs), is the potential outcomes (PO) framework. However, while the two approaches dedicate more or less emphasis to different aspects of causal inference, it has been argued that the SCM framework "does not exclude any concept or relationship definable in the PO approach" (see http://causality.cs.ucla.edu/blog/index.php/2020/01/29/on-imbens-comparison-of-two-approaches-to-empirical-economics/). Moreover, the SCM framework was developed by AI researchers (as opposed to the PO framework, which was mostly developed in the context of economics): it therefore puts particular emphasis on algorithmic aspects of causal reasoning (see, e.g., https://ftp.cs.ucla.edu/pub/stat_ser/r360.pdf for counterfactuals). This makes it particularly suited for our objective, where we want to algorithmically generate ground truth answers to causal questions, without having to assess the correctness of an answer based on common sense. We will be happy to elaborate on the aspects above in the Introduction, to better contextualize our work.
> > >
> > > In case the above does not answer your questions, maybe you could further let us know what you dislike about using the Pearl framework here? Why do you think it would be intellectually preferable not to use it?
> > >
> > >
> > > `Table 2: Can you explain why CausalCoT makes performance drop on Rung 3:` We agree that the result is in some sense counterintuitive: it is an empirical finding, and our explanation is that it might be related to difficulties in distinguishing the query type, as our investigation through CausalCoT shows, since this sub-task is harder for counterfactuals (rung 3) than for lower rungs, due to a broader range of queries to be considered. We can include this in the Limitations section too, in that a precondition for CoT to work well is that each subquestion can be well-addressed by LLMs, which is a bit difficult for the case of the query type questions in Rung 3. And for the significance test, yes we can run a significance test and add it to the next version of the paper.
> > >
> > >
> > > `one important step towards augmenting LLMs is understanding to what extent they already solve certain reasoning tasks.`
> > > We would agree that investigating the performance with plugins would be great, and it would be an extremely interesting aspect to investigate in the future. As you suggested, we will  discuss this future research direction in the paper. There is, as you mentioned, a thread of work attempting to improve LLM math abilities, e.g., https://arxiv.org/abs/2308.05713, where it is suggested that “the plugins significantly enhance GPT’s ability to solve these problems”. The problem is not solved though: as the authors remark, "there are still often ”interface” failures; that is, GPT often has trouble formulating problems in a way that elicits useful answers from the plug-ins". We expect that something similar will happen for causal inference, once suitable plugins are built, where the language-to-tool interface will still be a non-trivial research question. Moreover, the possible future existence of such plugins will make it even more important to have systematic benchmarks that assay causal inference capabilities in a broad set of tasks (not just correlation vs. causation), so we would view this as nicely complementary to what we do.

---

> > > > ### Comment · Reviewer_5uc1 · 2023-08-18
> > > > **Score change**
> > > >
> > > > Thank you so much for the detailed replies.
> > > >
> > > > **Regarding motivation for Pearl's framework**: I think it is fine to set on that framework, concretely I would suggest expanding the intro a bit in the lines that lead to motivate the problem you decided to study.  Regarding Pearl's framework, you could also point out its strengths and cite one or two alternatives if you like. The comparison to Kiciman was intended more to suggest improving the presentation that leads to motivate and *sell* the problem as important to the readers. This could also help improve citation chances as concurrent works will compete to be cited or built upon.
> > > >
> > > > **Regarding drop to GPT-4 on Rung 3**: Please do discuss it somewhere, I think it is important to emphasize if one wants to use the technique on Rung 3 problems. I don't think it's necessary to add detailed significance tests, maybe you could point out the ball-park of what type of difference scores are significant (maybe 1 or 2 accuracy points?).
> > > >
> > > > **Regarding tools**: Thanks for be willing to discuss it.
> > > >
> > > > I will raise the score from 4 to 5. I did not raise it to 6 because of the drop on Rung 3 compared to GPT-4.

---

> > > > > ### Author Response · Authors · 2023-08-18
> > > > > **Acknowledgement**
> > > > >
> > > > > Thank you very much for raising the score to acceptance, and all your time and efforts engaging in our discussions. We appreciate it.

---

### Official Review · Reviewer_bMQ8 · 2023-07-16

**Soundness:** 3 good
**Presentation:** 3 good
**Contribution:** 3 good
**Rating:** 6
**Confidence:** 4

**Summary:**

This paper evaluates the causal reasoning ability of language models and proposes a structured prompting approach to improve this ability. For this goal, the authors create a benchmark. The benchmark includes synthesized causal graphs and related queries. The answers to queries are computed based on a causal reasoning formalism given the causal graph.  The graph and queries are verbalized for creating the corpus to provide natural language context and questions.  For evaluation purposes, a variety of current large language models such as GPT 3.5 and GPT 4 are used.  The language models are given the questions that need causal reasoning to be answered.  In addition to the new benchmark and the LLM evaluations, the authors propose a structured chain of thought prompting strategy (called CausalCOT) to improve the ability of LLMs in causal reasoning. The idea is to provide step-by-step prompts based on the steps of formal reasoning. The steps include prompting for extraction of the causal graph,  classifying the query type, and several others.
The results show that structured prompting improves the causal reasoning ability of the large language models. Especially for the questions that are anti-commonsensical and are not contained training large language models, the models show a more considerable extent of improvements compared to the baselines.


**Strengths:**

The paper focuses on an important problem which is causal reasoning with LLMs.
The created benchmark is potentially a valuable evaluation resource.
Structured prompting is a recent research trend for improving LLMs, and the authors follow the same trend but with a novel structure that is based on causal reasoning steps.
The findings show the effectiveness of structured prompting that is in line with other related research.



**Weaknesses:**

--It is not clear to me how the LLMs follow each step of reasoning:  Is this totally zero-shot? or do you do in-context learning with a few examples? if this is zero-shot, it is a bit hard to see why the model knows about causal graphs and specific query types for example.
--What is the lexicon used for verbalization of the causal model? How many distinct works are used? How do you evaluate if the text  that is generated based on a template, is in line with commonsense or not?
--The data set is synthetic which is fine. However, do you have any realistic scenarios to test the model on? I mean some realistic problem setting that shows this causal reasoning is required and the structured prompting is helpful in that realistic setting. More specifically, do we have real natural language explanations that explain complex casual graphs? can you add more discussions in this direction to the paper?

**Questions:**

See the above weakness.

**Limitations:**

there is a section on this in the paper.

---

> ### Author Rebuttal · Authors · 2023-08-10
>
> We appreciate the reviewer’s comment that this work “focuses on an important problem”, “The created benchmark is potentially a valuable evaluation resource”, and our CausalCoT is “a novel structure that is based on causal reasoning steps”.
>
> > Is this totally zero-shot? or do you do in-context learning with a few examples?
>
> To avoid effects of arbitrary choices of hyperparameters of a few-shot evaluation (e.g. number of samples and sampling method), we report zero-shot performance of all models in the main results (Section 5.2), since most LLMs are trained on the large internet corpus that covers basic causality knowledge. In the paper, we also did a small-scale analysis of the few-shot setting in Sec 5.6 “Effect of In-Context Learning”.
>
> During the rebuttal period, we extended the experiments, and obtained the performance for few-shot learning using 10 examples to instruct CausalCoT, leading to an accuracy of 70.09% (+4 point increase over the 0-shot CoT in Table 2), as the model learns more correct reasoning from the examples.
>
> > if this is zero-shot, it is a bit hard to see why the model knows about causal graphs and specific query types for example.
>
> As shown in Figure 1, the first step is causal relation extraction (i.e., text-to-causal graph parsing). In each input prompt, all necessary the causal relations are explicitly mentioned (in natural language), so instead of inferring causal relations from training data, the model only has to parse the input and then solve the causal inference problem. This way we focus our evaluation on formal reasoning, rather than commonsense-based causal reasoning as in related work.
>
> Hence, in the error analysis by subquestions (Section 5.4), the first step tests whether models can comprehend causal text expressions well enough to ground them into symbolic forms. And the second step to classify a question into specific query types is also a skill that LLMs should have knowledge of given its training data of the entire internet and many book corpora. For example, GPT-3.5 and GPT-4 can act as a causality textbook to explain the definition of each query type and give examples. And this step is to test the reverse ability. For example, the question “Is the chance of YY larger when observing XX?” is asking about correlations (Rung 1), and the question “Will XX increase the chance of YY?” is asking about average treatment effect (Rung 2). We regard such skills within the current capabilities of LLMs given its large training data.
>
> > How many distinct works are used?
>
> As listed in Appendix A.1, we collected causal questions from a list of nine commonly-used causality books and papers (Pearl et al., 2000; Glymour et al., 2016; Peters et al., 2017; Pearl and Mackenzie, 2018; Neal, 2020; Halpern and Pearl, 2005a; Halpern and Pearl, 2005b; Hopkins and Pearl, 2007; Pearl, 2009a).
>
> > What is the lexicon used for verbalization of the causal model?
>
> We report the question statistics in Table 1, where we have on average 94.47 words per question, and 539 vocabulary words in total. We also show data examples in Appendix Tables 5-12.
>
> > How do you evaluate if the text that is generated based on a template, is in line with commonsense or not?
>
> As in lines 171-183, we collect commonsensical causal stories from common examples in causal inference books and papers (in Appendix A.1), which always set up the problems in a realistic way, such as the drug-gender-recovery example of Pearl and Mackenzie (2018) to illustrate Simpson’s paradox. And for the non-commonsensical stories, their detailed generation process are in line 177-183.
>
> > The data set is synthetic which is fine. However, do you have any realistic scenarios to test the model on? I mean some realistic problem setting that shows this causal reasoning is required and the structured prompting is helpful in that realistic setting. More specifically, do we have real natural language explanations that explain complex causal graphs? Can you add more discussions in this direction to the paper?
>
> Yes, we can add more discussions about extending the dataset to more real-world scenarios in the camera-ready version of the paper, such as fake news debunking, and logical fallacy correction. In the meantime, when composing the dataset’s commonsensical part, we took care to set many problems more realistically. Some of the “commonsensical” questions included in our dataset were inspired by real-world examples of policy-relevant causal inference questions. For example, the dataset includes questions on vaccine efficacy (see the example in our Figure 1) inspired by similar questions which arose in the context of the COVID-19 pandemic, and where incorrect causal reasoning resulted in fallacies where vaccinations were considered be harmful instead of beneficial in preventing severe cases (see, e.g., https://www.covid-datascience.com/post/israeli-data-how-can-efficacy-vs-severe-disease-be-strong-when-60-of-hospitalized-are-vaccinated).
>
> Another note for the general usage of our theoretical framework is that the notions of NDE and NIE are very applicable to cases like bias and discrimination. Let’s take the example of how the double-blind review process for papers work: Potentially, we have the causal graph of author identity->paper quality->acceptance and also author identity->acceptance. The unfair causal effect is the natural direct effect, i.e., NDE(author identity->acceptance), and that’s why author identities are anonymized to avoid this problem.

---

> > ### Comment · Reviewer_bMQ8 · 2023-08-16
> >
> > I have read the reviews and the author's response. I was already positive about this work but I also see the limitations in verifying the validity of the generated model and the extension to the realistic domain. My score stays unchanged.

---

> > > ### Author Response · Authors · 2023-08-17
> > >
> > > Thank you very much for your positive feedback of our work. Appreciate it!

---

### Author Rebuttal · Authors · 2023-08-10

Firstly, we would like to thank all reviewers for a wealth of feedback. Four out of five reviewers recommended acceptance, and we are encouraged by the large number of thoughtful comments and suggestions, which demonstrate not only that all reviewers largely understood our focus, but also that they expressed interest in our work.

We appreciate the acknowledgement of our work’s contributions in the following ways:
1. **Relevance:** This work “focuses on an important problem” (bMQ8) and “provides a way to formally evaluate the causal reasoning ability of LLMs for the first time” (hcc4).
1. **Dataset contribution:** The created benchmark is “the first formal causal inference/reasoning dataset” (hcc4), thus “a valuable evaluation resource” (bMQ8) and “The paper nicely builds on the framework proposed by Pearl with a dataset of 10K questions covering a large variety of causal graph structures and estimands” (rbbm).
1. **Method contribution:** Our CausalCoT is “a novel structure that is based on causal reasoning steps” (bMQ8) and “which lets LMs explicitly state causal problems and then solve [them]” (YWrR).
1. **Comprehensive evaluation results:** “The evaluation of state of the art models on the task is well conducted with a particular methodology to isolate the effect of data contamination.” (rbbm) and addressing "effects of memorization in LLMs" (YWrR).

A shared question across most reviewers is more clarification of whether we used a zero-shot or few-shot evaluation. We clarified in our response that our main experiments are zero-shot to simplify analysis, while we include some few-shot results in the paper. Upon the request of the reviewers, we have also conducted several new experimentsto report the few-shot performance of all models. Additionally, some reviewers asked about the quality of the dataset, for which we conduct new human evaluations and automatic evaluations to gain better insight into the quality of our data. Finally, we also supplemented our rebuttal with ablation studies and various experiments requested by the reviewers (discussed in the corresponding rebuttals).

We look forward to the discussions to make any further clarifications to help the reviewers reach a consensus on our submission's value to the community.

---

### Decision · Program_Chairs · 2023-09-21

**Decision:**

Accept (poster)

**Comment:**

There is consensus between reviewers that this paper presents a solid contribution in the form of a causal reasoning benchmark that can be of large importance to the community.  There is excellent discussion between authors and reviewers regarding some subset of topics presented in the paper.  Given the excitement about the paper, I suggest that we accept it to the conference.